# UNRESTRAINED SIMPLEX DENOISING
# FOR DISCRETE DATA
# A NON-MARKOVIAN APPROACH
# APPLIED TO GRAPH GENERATION

## ABSTRACT

Denoising models such as Diffusion or Flow Matching have recently advanced generative modeling for discrete structures, yet most approaches either operate directly in the discrete state space, causing abrupt state changes. We introduce simplex denoising, a simple yet effective generative framework that operates on the probability simplex. The key idea is a non-Markovian noising scheme in which, for a given clean data point, noisy representations at different times are conditionally independent. While preserving the theoretical guarantees of denoising-based generative models, our method removes unnecessary constraints, thereby improving performance and simplifying the formulation. Empirically, *unrestrained simplex denoising* surpasses strong discrete diffusion and flow-matching baselines across synthetic and real-world graph benchmarks. These results highlight the probability simplex as an effective framework for discrete generative modeling.

## 1 INTRODUCTION

Denoising models, such as Discrete Diffusion and Discrete Flow Matching, have substantially advanced generative modeling for discrete structures (Austin et al., 2021; Campbell et al., 2022; 2024; Gat et al., 2024), including graphs (Haefeli et al., 2022; Vignac et al., 2023; Qin et al., 2025; Boget, 2025). Yet, surprisingly few works study denoising on simplices for modeling discrete data (Richemond et al., 2022; Avdeyev et al., 2023; Stark et al., 2024). In graph generation specifically, existing generative models do not extend simplex-based parameterizations beyond the 1-simplex (Liu et al., 2025).

Discrete diffusion models such as Austin et al. (2021) and Campbell et al. (2022) impose transitions between discrete states during denoising. Such jumps can abruptly alter the nature of a noisy instance, for example, disconnecting a graph by deleting an edge. By allowing probabilistic superpositions of categories, denoising on the simplex enables smooth transitions between states.

However, naïve noising schemes on the simplex exhibit undesirable behavior. In particular, avoiding abrupt discontinuities in the support of the noisy distributions requires a carefully designed probability path. To mitigate these issues, prior simplex-based denoising methods (e.g., Richemond et al., 2022; Stark et al., 2024) introduce non-trivial theoretical formulations, which may hinder broader adoption.

Moreover, standard approaches such as diffusion or flow matching strongly constrain the reverse dynamics by enforcing a continuous denoising trajectory in the continuous-time limit, i.e., $\lim_{dt \to 0} p_{t+dt|t}(\mathbf{x}_{t+dt} \mid \mathbf{x}_t) = \delta(\mathbf{x}_{t+dt} - \mathbf{x}_t)$. In discrete diffusion, this constraint has been linked to compounding denoising errors (Lezama et al., 2023; Boget, 2025; Zhang et al., 2025). We argue that similar dynamics operate in the continuous case.

In this work, we introduce *simplex denoising*, a simple yet effective generative model for discrete data that operates directly on the probability simplex. Our model leverages a non-Markovian diffusion framework that assumes independence across noisy states (conditional on the clean input).

By removing dependence on the previous noisy state, our formulation eliminates unnecessary constraints and simplifies the denoising process.

Our key contributions are as follows:

- We introduce a new non-Markovian denoising approach for categorical data operating on the probability simplex (Section 4). The formulation efficiently substantially simplifies standard denoising paradigms such as diffusion or flow matching and addresses the compounding denoising errors issue.

- We provide theoretical guarantees, proving convergence of the proposed procedure under the true denoising distribution $p_{1|t}(\mathbf{x} \mid \mathbf{x}_t)$ (Section 4).

- Leveraging explicit probability paths, we propose new theoretical tools for the analysis of the noising dynamics on the probability simplex. Specifically, we introduce the Voronoi probability and its closed-form formula for the Dirichlet probability path (Section 3).

- Empirically, our unrestrained simplex denoising method surpasses strong discrete diffusion and flow-matching baselines on synthetic and real-world graph benchmarks, advancing the state of the art on multiple datasets (Section 5).

## 2 BACKGROUND

In this paper, we focus on modeling multivariate categorical data, with application to graphs that may have discrete node and edge attributes. In this section, we begin by establishing some notations used throughout the paper. We then position our contribution within the literature along three axes: (i) denoising-based generative models for discrete data; (ii) non-Markovian formulations of noising and denoising processes; and (iii) generative models for graphs. We conclude by motivating our choice to operate directly on the probability simplex.

### 2.1 NOTATIONS

We define a graph as a set of nodes and edges, denoted by $\mathcal{G} = (\mathcal{V}, \mathcal{E})$. We represent each node pair $(\nu_i, \nu_j)$ as a vector $\boldsymbol{a}_{i,j} \in \mathbb{R}^{d_a}$, where $d_a$ is the number of edge types, including the absence of an edge. Similarly, we represent the node attributes as a vector $\boldsymbol{v}_i \in \mathbb{R}$, but we do not need to encode the absence of a node. When the formulation applies unambiguously to node and edge representations, we use $\boldsymbol{x}$ to refer to both $\boldsymbol{v}^{(i)}$ and $\boldsymbol{a}^{(i,j)}$ to facilitate readability. We denote $\boldsymbol{e}_i$ as the $i$th standard basis vector.

We adopt the flow matching convention $t \in [0, 1]$, with $t = 1$ corresponding to the data distribution $\mathbf{x}_1 \sim q_{\text{data}}(\mathbf{x})$ and $t = 0$ to the noise distribution $\mathbf{x}_0 \sim q_0(\mathbf{x})$. We denote $dt$ as a small but not necessarily infinitesimal time interval such that $t + dt \leq 1$.

### 2.2 DENOISING MODELS FOR DISCRETE DATA

We group existing approaches for discrete data into three broad families: (1) continuous relaxations, (2) discrete noising, and (3) simplex denoising.

Approaches based on *continuous relaxation* embed discrete variables into an unconstrained Euclidean space and train conventional continuous diffusion models with, e.g., a standard Gaussian prior (Jo et al., 2022; 2024; Niu et al., 2020). By encoding unordered categories into $\mathbb{R}$, these methods create artefactual orderings and distances, which may bias generation.

Initiated by D3PM (Austin et al., 2021), *discrete diffusion* corrupts data by injecting discrete noise and models transitions directly in the discrete domain. Subsequent work extends this idea to continuous time (Campbell et al., 2022) and to discrete flow matching (Campbell et al., 2024; Gat et al., 2024). In graph generation, DiGress (Vignac et al., 2023) and DeFog (Qin et al., 2025) exemplify this line of work. By jumping between discrete states during denoising, these models may alter the nature of the instance abruptly and drastically. For instance, removing one edge may disconnect a graph.

In *simplex denoising*, categorical variables are represented as points on the probability simplex (vertices for one-hot encodings). One strategy diffuses in $\mathbb{R}^K$ and leverages distributional correspondences, such as Gamma–Dirichlet or normal–logistic–normal, to map noisy samples back onto the simplex (Richemond et al., 2022; Floto et al., 2023). A second strategy generates noise *directly* on the simplex (Avdeyev et al., 2023; Stark et al., 2024). In this work, we adopt the latter approach.

### 2.3 NON-MARKOVIAN DIFFUSION

In statistics and physics, the strict definition of a diffusion process entails the Markov property. In deep generative modeling, however, "diffusion" is used more broadly to denote a variety of forward noising schemes. By defining the forward kernel $q_{t|t+dt,1}$ to depend on both the preceding noisy state and the clean sample, DDIM (Song et al., 2021) is, to our knowledge, the first *non*-Markovian diffusion model.

Recently, a family of non-Markovian formulations specifies the noising process uniquely as a function of the noise level $\alpha_t$ and the clean data $x_1$, making $q_{t|1}(x_t \mid x_1)$ independent of all other noisy states (Boget, 2025; Chen et al., 2025; Zhang et al., 2025). During denoising, this independence assumption removes direct dependencies on previous noisy states and thereby mitigates error accumulation ("compounding denoising errors") (Lezama et al., 2023). Our work is, to the best of our knowledge, the first to adapt this class of non-Markovian methods to simplex-based denoising.

### 2.4 GRAPH GENERATIVE MODELS

Recently, *denoising models* coupled with equivariant GNNs have become the dominant approach, as they have been shown to be highly effective (for other approaches, see Appendix C.1). Continuous-space methods include score-based diffusion (Yang et al., 2019; Jo et al., 2022), diffusion bridges (Jo et al., 2024), and flow matching (Eijkelboom et al., 2024). Discrete formulations such as diffusion in discrete time (Haefeli et al., 2022; Vignac et al., 2023) or in continuous time (Xu et al., 2024), flow matching (Qin et al., 2025), and non-Markovian discrete diffusion (Boget, 2025), also report strong results. However, these approaches do not perform *smooth* diffusion on the probability simplex: they either noise graphs in an unconstrained Euclidean space or operate as jump processes over discrete states.

Beta diffusion (Liu et al., 2025) applies continuous noise on the 1-simplex via the Beta distribution. An extension to the $K$-dimensional probability simplex might appear straightforward. However, no model achieves this extension directly, indicating that this is more challenging than it appears. Liu et al. (2025) generalize their approach by encoding $K$-categorical variables as $K$ binary variables. Eijkelboom et al. (2024) adopt the Dirichlet flow matching formulation but report poor performance, leading them to relax the noising process to $\mathbb{R}^K$. Recently, Graph Bayesian Flow Network (Song et al., 2025) proposes a diffusion mechanism that adds Gaussian noise in $\mathbb{R}^K$ followed by a projection into the simplex. Sampling proceeds via a dual update scheme in which the update type is selected based on the Euclidean progress achieved by the previous step. Taken together, these developments suggest that simplex-based denoising is more difficult than one might expect. We address this challenge by *unrestraining* the denoising process, yielding a simple yet effective model that operates *directly* on the probability simplex. In the next section, we further motivate our approach.

### 2.5 RATIONALE FOR DENOISING ON THE SIMPLEX

We motivate denoising directly on the probability simplex by contrasting it with discrete denoising approaches and methods that operate in $\mathbb{R}^K$.

Discrete and continuous noising differ fundamentally in the structure of the corrupted data. In discrete schemes, noisy samples remain sparse, whereas continuous perturbations produce fully connected weighted graphs. Continuous noising avoids discontinuities, most notably graph disconnection, and yields edge weights that naturally encode proximity, providing each node with information about its relative position to all others. More generally, continuous vectors are richer than one-hot representations of the same dimension and supply unique identifiers for nodes and edges, thereby alleviating the expressivity limitations of GNNs (Loukas, 2020). This prevent the need for costly, hand-crafted structural features typically required in discrete diffusion and flow matching methods (see Appendix C.2). Importantly, the sparsity of discrete noising does not translate into sparse GNN

computation. Edge representations and probabilities must still be evaluated for all node pairs. Thus, the dense complexity remains.

Modeling categorical distributions in $\mathbb{R}^K$ introduces limitations that are avoided by operating directly on the probability simplex. First, points on the simplex have only $K-1$ degrees of freedom; embedding them in $\mathbb{R}^K$ adds a redundant dimension that carries no information and unnecessarily complicates the dynamics. Second, simplex-based noising naturally defines explicit and interpretable probability paths over categorical distributions (Section 3), whereas Euclidean noising requires nonlinear projection or renormalization steps whose behavior is harder to characterize. Finally, points on the simplex admit a direct probabilistic interpretation, which is both conceptually elegant and facilitates the design of the denoising process (Section 4.2).

## 3 NOISING ON THE SIMPLEX

Noising processes defined on the probability simplex pose specific challenges. In particular, specifying $q_t$ as an *interpolant* between endpoint distributions $q_1$ and $q_0$, as is common in discrete diffusion and flow matching, restricts the support of $q_t$, leaving large regions of $\mathbb{S}_K$ out of distribution and creating jump discontinuities in $q_t$. In this section, we formalize the setting, introduce an explicit, well-behaved parameterization of the noising path, and propose new theoretical tools for the analysis of the noising dynamics.

### 3.1 PROBLEM SETTING

Let $\mathbb{S}_K$ denote the $K$-1-dimensional probability simplex, $\mathbb{S}_K = \left\{ \boldsymbol{x} \in \mathbb{R}^K \,\middle|\, \mathbf{1}_K^\top \boldsymbol{x} = 1, \ x_i > 0 \ \forall i \right\}$. We represent categorical node/edge attributes as vertices of the simplex, i.e., $\mathbf{x}_1 \in \left\{ \boldsymbol{e}_i \right\}_{i \in [K]}$.

A noising process on the simplex is a family of conditional distributions $(q_t)_{t \in [0,1)}$ with $q_t(\cdot \mid \mathbf{x}_1)$ supported on $\mathbb{S}_K$ for all $t \in [0,1)$, and boundary conditions $\lim_{t \to 1} q_t(\mathbf{x}_t \mid \mathbf{x}_1) = \delta(\mathbf{x}_t - \mathbf{x}_1)$, $q_0(\mathbf{x}_t \mid \mathbf{x}_1) = q_0(\mathbf{x}_t)$, i.e., the terminal distribution collapses to the clean vertex and the initial distribution is independent of $\mathbf{x}_1$. Such processes can be specified either implicitly via a stochastic interpolant or explicitly by choosing a parametric family on the simplex.

The interpolant construction is standard in discrete diffusion and flow matching, and also underlies *Simple Iterative Denoising*, which can be viewed as the discrete analogue of our approach.. It sets $\mathbf{x}_t = \alpha_t \mathbf{x}_1 + (1 - \alpha_t)\mathbf{x}_0$, with $\mathbf{x}_1 \sim p_{\text{data}}$, $\mathbf{x}_0 \sim q_0$, and $\alpha_t \in [0,1]$. However, this formulation exhibits what Stark et al. (2024) call pathological properties. In particular, within the interpolant framework, the posterior $p_{1|t}(\mathbf{x}_1 \mid \mathbf{x}_t) \propto p_{\text{data}}(\mathbf{x}_1)q_t(\mathbf{x}_t \mid \mathbf{x}_1)$ has support on at most $K-1$ vertices for all noise levels $\alpha_t > 1/K$, so that the posterior is trivial for $\alpha_t > 0.5$. For all $t$, $q_t(\mathbf{x}_t \mid \mathbf{x}_1)$ is a uniform distribution over subsets of $\mathbb{S}_K$, and zero elsewhere, producing support discontinuities. Figure 1 illustrates such a process on the 2-simplex. Such piecewise-constant behavior is ill-suited for approximating posteriors with differentiable neural networks. We avoid these pathologies by specifying an explicit, parametric probability path on the simplex.

Dirichlet distributions (and their mixtures) offer a natural, tractable choice on $\mathbb{S}_K$. Recall the Dirichlet density function: $\text{Dir}(\boldsymbol{x}; \boldsymbol{\alpha}) = \text{B}(\boldsymbol{\alpha})^{-1} \prod_{i=1}^K x_i^{\alpha_i - 1}$, where $\text{B}$ is the multivariate Beta function (see Equation 10). For instance, a simple yet useful noising path is defined as $q_t(\mathbf{x}_t \mid \mathbf{x}_1) = \text{Dir}\big(\mathbf{x}_t; \mathbf{1} + \alpha_t \mathbf{x}_1\big)$, with a non-decreasing noise schedule $\alpha_t \in \mathbb{R}_{\geq 0}$ such that $\alpha_0 = 0$ and $\lim_{t \uparrow 1} \alpha_t = \infty$, e.g., $\alpha_t = -a \log(1-t)$, where $a$ is a hyperparameter. This way, $q_t$ has full support on $\mathbb{S}_K$ for any $t$, thereby avoiding the support-collapse issues of interpolant-based methods. Figure 1 illustrates noising based on a parametric probability path.

### 3.2 VORONOI PROBABILITIES

The Voronoi regions of the simplex vertices provide an intuitive lens on how a noising process behaves. Let $\mathbb{V}_k \subseteq \mathbb{S}_K$ denote the Voronoi region of vertex $k$ such that $\mathbb{V}_k = \{\boldsymbol{x} \in \mathbb{S}_K \mid d(\boldsymbol{x}, \boldsymbol{e}_k) \leq d(\boldsymbol{x}, \boldsymbol{e}_j) \ \forall j \neq k\}$, with $d(\cdot, \cdot)$ the Euclidean distance. Geometrically, $\mathbb{V}_k$ is the polytope of points whose nearest simplex vertex is $\boldsymbol{e}_k$. Equivalently, we have $\mathbf{x} \in \mathbb{V}_k \iff x_k = \max_j x_j$.

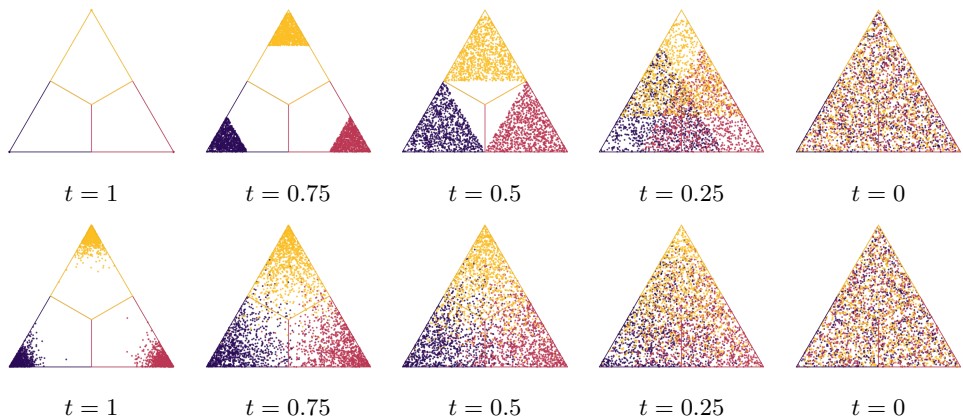

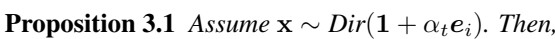

Figure 1: Upper row: Noising with linear interpolant creates discontinuities and, for $t > 0.5$, all points remain in their Voronoi region. Lower row: Noising with explicit parametric probability path avoids discontinuities.

Given the conditional noising distribution $q_t(\mathbf{x}_t \mid \mathbf{x}_1)$, we define the *Voronoi probability* for vertex $k$ as the conditional probability $P_{v_k}(\mathbf{x}_t \in \mathbb{V}_k \mid \mathbf{x}_1 = \boldsymbol{e}_k)$, i.e., the probability that the closest vertex to $\mathbf{x}_t$ remains its origin $\mathbf{x}_1$. Intuitively, this quantity summarizes how quickly the unconditional distribution $q_t(\mathbf{x})$ "mixes".

For the stochastic-interpolant construction, for instance, one has $P(\mathbf{x}_t \in \mathbb{V}_i \mid \mathbf{x}_1 = \boldsymbol{e}_i) = 1$ for all $t$ such that $\alpha_t > 0.5$; and conversely, the posterior over vertices collapses to the Voronoi indicator $P_{1|t}(\mathbf{x}_1 = k \mid \mathbf{x}_t) = [\mathbf{x}_t \in \mathbb{V}_k]$.

For the probability path $\mathrm{Dir}(\mathbf{x}_t; \mathbf{1} + \alpha_t \mathbf{x}_1)$ introduced above, the Voronoi probability admits a closed-form expression:

**Proposition 3.1** *Assume* $\mathbf{x} \sim Dir(\mathbf{1} + \alpha_t \boldsymbol{e}_i)$. *Then,*

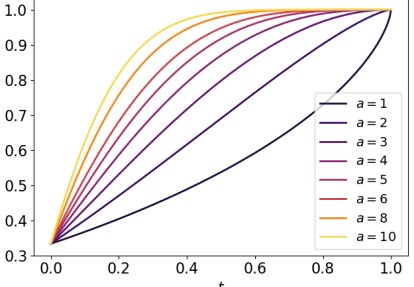

Figure 2: Voronoi probabilities over time for $\mathbf{x}_t \in \mathbb{S}_3 \sim \mathrm{Dir}(\mathbf{1} + \alpha_t \mathbf{x}_1)$, with $\alpha_t = -a \log(1-t)$ for various values of $a$. For $a = 1$, $P_{v_k}$ increases rapidly as $t \to 1$; for $a = 10$, $P_{v_k}$ is already close to 1 by $t = 0.6$. Suitable choices of $a$ therefore likely lie between 1 and 10.

$$P_{v_i}(\mathbf{x} \mid \boldsymbol{e}_i) = \sum_{k=0}^{K-1} \frac{(-1)^k \binom{K-1}{k}}{(k+1)^{\alpha_t - 1}} \tag{1}$$

All proofs are presented in Appendix A.

We leverage Voronoi probabilities to design and calibrate explicit probability paths. Figure 2 illustrates how the Voronoi probabilities evolve over time for various values of $a$. In Appendix E.2.2, we evaluate our model across multiple values of $a$, demonstrating the usefulness of Voronoi probabilities for calibrating the noise scheduler.

## 4 METHOD

In this section, we present our *Unrestrained Simplex Denoising* (UNSIDE) method. As with any other denoising model, it includes a noising process that progressively corrupts the data, and a learned denoising backward process. The multivariate forward (noising) process acts independently on each dimension $\boldsymbol{x}^{(i)}$, $i \in [L]$, noising instances on the multi-simplex $\mathbb{S}_K^L$, where $L$ is the number of dimensions. In contrast, during the reverse process used at inference, dependencies across dimensions are captured by a learned denoiser that outputs element-wise logits conditioned on all dimensions $\mathbf{x}_t^{(1:L)}$. This setup follows prior work on discrete diffusion and flow matching (Austin et al.,

2021; Vignac et al., 2023; Campbell et al., 2024; Stark et al., 2024; Boget, 2025). Our UNSIDE framework can be viewed as a continuous analogue of Simple Iterative Denoising (SID) (Boget, 2025). Importantly, extending SID to the continuous setting, much like adapting diffusion or flow matching frameworks across data modalities, requires substantial methodological modifications and represents a significant contribution of this work. A detailed comparison between UNSIDE and SID is provided in Appendix C.3.

### 4.1 NOISING

Our noising mechanism relies on two key components: (i) a probability path that governs corruption at any continuous time $t \in [0, 1]$; and (ii) a conditional-independence assumption across different times.

**Probability path** We define a probability path $(q_t)_{t \in [0,1)}$ with boundary conditions $\lim_{t \to 1} q_t(\mathbf{x} \mid \mathbf{x}_1) = \delta(\mathbf{x} - \mathbf{x}_1)$ and $q_0(\mathbf{x} \mid \mathbf{x}_1) = q_0(\mathbf{x})$. Since the forward process operates independently across dimensions, the distribution at time $t$ factorizes as

$$q_t\big(\mathbf{x}^{(1:L)} \mid \mathbf{x}_1^{(1:L)}\big) = \prod_{i=1}^{L} q_t\big(\mathbf{x}^{(i)} \mid \mathbf{x}_1^{(i)}\big). \tag{2}$$

In principle, our method accommodates any path $(q_t)_{t \in [0,1)}$. For efficient training, however, we require (a) the ability to easily sample from $q_t$ for all $t$, and (b) well-behaved noise distributions as described in Section 3. We further discuss the choice of the probability path in the context of graph generation in Section 4.6. In practice, Dirichlet distributions (or mixtures thereof) provide a convenient choice, e.g., $q_t(\mathbf{x} \mid \mathbf{x}_1) = \mathrm{Dir}(\mathbf{1} + \alpha(t)\mathbf{x}_1)$, where $\alpha \in \mathbb{R}_{\geq 0}$ is an increasing function with $\alpha_0 = 0$ and $\lim_{t \to 1} \alpha_t = \infty$.

**Independence across time** Unlike standard diffusion models such as DDPM, DDIM, or D3PM, we do not define direct dependencies between $\mathbf{x}_t$ and $\mathbf{x}_{t+dt}$. Instead, we assume conditional independence of intermediate noisy states given the clean input, as formalized in the following assumption:

**Assumption 4.1** *For any $s \neq t$, we assume the following conditional-independence property:*
$$q_t(\mathbf{x}_t \mid \mathbf{x}_s, \mathbf{x}_1) = q_t(\mathbf{x}_t \mid \mathbf{x}_1). \tag{3}$$

Because training typically draws a single noisy sample at a randomly chosen time $t$, this assumption has no practical effect on the training objective. It does, however, have major implications for the denoising process.

### 4.2 DENOISING

Let $P_{1|t}(\mathbf{x}_1^{(i)} \mid \mathbf{x}_t^{(1:L)})$ denote the posterior distribution of a clean element given a noisy instance at time $t$. Since this posterior $P_{1|t}(\mathbf{x}_1^{(i)} \mid \mathbf{x}_t^{(1:L)}) = \mathrm{Cat}(\mathbf{x}; \boldsymbol{\pi})$ is a categorical distribution, it is fully specified by its parameter $\boldsymbol{\pi}$. We denote $p_{t+dt}(\mathbf{x}_{t+dt}^{(i)} \mid \mathbf{x}_t^{(1:L)})$ a single denoising transition and obtain the following result.

**Proposition 4.2** *Given $P_{1|t}(\mathbf{x}_1^{(i)} \mid \mathbf{x}_t^{(1:L)})$ and Assumption 4.1, the one-step denoising kernel satisfies*

$$p_{t+dt}(\mathbf{x}_{t+dt}^{(i)} \mid \mathbf{x}_t^{(1:L)}) = \sum_{\mathbf{x}_1^{(i)}} q_{t+dt}(\mathbf{x}_{t+dt}^{(i)} \mid \mathbf{x}_1^{(i)}) P_{1|t}(\mathbf{x}_1^{(i)} \mid \mathbf{x}_t^{(1:L)}) \tag{4}$$

$$= \mathbb{E}_{\mathbf{x}_1^{(i)} \sim P_{1|t}(\cdot \mid \mathbf{x}_t^{(1:L)})} \left[ q_{t+dt}(\mathbf{x}_{t+dt}^{(i)} \mid \mathbf{x}_1^{(i)}) \right]. \tag{5}$$

Thus, each denoising step can be interpreted as (i) sampling a clean element $\mathbf{x}_1^{(i)}$ from the posterior $P_{1|t}(\mathbf{x}^{(i)} \mid \mathbf{x}_t^{(1:L)})$, followed by (ii) re-noising this sample via $q_{t+dt}(\cdot \mid \mathbf{x}_1^{(i)})$. Equivalently, we sample from the noisy distribution under the expected clean posterior. We emphasize that the simplicity of these formulations relies on the fact that $q_{t+dt}(\mathbf{x}_{t+dt}^{(i)} \mid \mathbf{x}_1^{(i)})$ remains on the simplex.

At inference time, we fix the number of function evaluations (NFE) to $T$ and use a uniform step size $dt = 1/T$. The full reverse procedure consists of iterating $T$ times the transition in Equation 4.

### 4.3 Convergence and Corrector Sampling

For simplicity, we first consider the univariate case. By fixing the time index in Equation 4, we obtain the Markov kernel $p_{t'|t}(\mathbf{x}'_t \mid \mathbf{x}_t) = \sum_{\mathbf{x}_1^{(i)}} q_t(\mathbf{x}'_{t'} \mid \mathbf{x}_1) P_{1|t}(\mathbf{x}_1 \mid \mathbf{x}_t)$.

**Proposition 4.3** *Assume that $P_{1|t}(\mathbf{x}_1 \mid \mathbf{x}_t)$ and $q(\mathbf{x}_t \mid \mathbf{x}_1)$ have full support on $[K]$ and $\mathbb{S}_K$, respectively. Further assume that the conditional independence holds: $q_t(\mathbf{x}'_t \mid \mathbf{x}_t, \mathbf{x}_1) = q_t(\mathbf{x}'_t \mid \mathbf{x}_1)$. Then the Markov kernel $p_{t'|t}(\mathbf{x}'_t \mid \mathbf{x}_t) = \sum_{\mathbf{x}_1 \in \mathcal{S}^L} P_{1|t}(\mathbf{x}_1 \mid \mathbf{x}_t) q_t(\mathbf{x}'_t \mid \mathbf{x}_1)$*

*converge to the stationary distribution*

$$\pi_t(\mathbf{x}_t) = \sum_{\mathbf{x}_1 \in \mathcal{S}^L} p(\mathbf{x}_1) q_t(\mathbf{x}_t \mid \mathbf{x}_1), \tag{6}$$

Fixing the time index during denoising therefore yields a simple *corrector* sampler, analogous in spirit to the corrector step of Gat et al. (2024) for discrete flow matching. The proof easily extend to the multivariate case (see Appendix A). When the true posterior $P_{1|t}(\mathbf{x}_1 \mid \mathbf{x}_t)$ is used, iteratively sampling from the kernel converges to $\pi_t$. Thus, iterating denoising and corrector steps to stationarity recovers samples from $p(\mathbf{x}_1)$. In practice, sufficiently small time steps $dt$ provide accurate approximations.

### 4.4 Advantages over Diffusion and Flow Matching

While our procedure resembles standard diffusion and flow matching, there is a key distinction. In classical reverse processes one typically enforces $\lim_{dt \to 0} p_{t+dt|t}(\mathbf{x} \mid \mathbf{x}_t) = \delta(\mathbf{x} - \mathbf{x}_t)$. By Assumption 4.1, our model does not impose this constraint. Intuitively, diffusion and flow matching update the sample by interpolating between the current noisy state and the denoiser's prediction, whereas our method directly resamples a less noisy state conditioned on the denoiser's prediction.

This observation provides intuition for the sampling benefits of our approach. In diffusion and flow matching, if $\mathbf{x}_t$ lies in a region where the learned denoiser $P_{1|t}^\theta$ poorly approximates the true posterior $P_{1|t}$, or in a region highly sensitive to small perturbations (e.g., near equilibrium points), then the next iterate $\mathbf{x}_{t+dt}$ remains close to $\mathbf{x}_t$, causing approximation errors to compound over time. In contrast, our method resamples from $P_{1|t}^\theta$ at each step, so $\mathbf{x}_{t+dt}$ need not stay in the same local region. This reduces error compounding by allowing the trajectory to escape regions that are poorly approximated and/or sensitive to small perturbations.

In addition, our formulation constitutes a substantial simplification relative to standard denoising methods such as diffusion and flow matching. Compared to diffusion, Assumption 4.1 yields the simplification $q_{t+dt}(\mathbf{x}_{t+dt} \mid \mathbf{x}_1, \mathbf{x}_t) = q_{t+dt}(\mathbf{x}_{t+dt} \mid \mathbf{x}_1)$. Compared to flow matching, it removes the need to construct a vector field that realizes the chosen probability path and to evaluate its derivatives at every denoising step, operations that are computationally expensive (See sampling time comparison in Appendix E.1). In contrast, our method is both simple and computationally efficient. We therefore regard this simplification as an important contribution.

### 4.5 Parametrization and Learning

As in discrete diffusion, the posterior $P_{1|t}(x_1 \mid \mathbf{x}_t^{(1:L)})$ is intractable. We approximate it with a neural network $f_\theta(\mathbf{x}_t^{(1:L)}, t) = \boldsymbol{\pi}^{(1:L)}$, referred to as the *denoiser*, which outputs element-wise logits for the categorical distribution. To respect graph symmetries, we instantiate $f_\theta$ as a Graph Neural Network (GNN), ensuring equivariance to node permutations.

Our training objective matches the $x$-prediction of discrete denoising and discrete flow matching methods, so any of their criteria can be adopted. Following Vignac et al. (2023), we minimize a weighted negative log-likelihood:

$$\mathcal{L}_\theta = \mathbb{E}\left[\gamma \sum_{\mathbf{v}_1}\left[-\log(p_\theta(\mathbf{v}_1 \mid \mathcal{G}_t))\right] + (1-\gamma)\sum_{\mathbf{a}_1}\left[-\log(p_\theta(\mathbf{a}_1 \mid \mathcal{G}_t))\right]\right],\tag{7}$$

where the expectation is over $t \sim \mathcal{U}(0,1)$, $\mathcal{G}_1 \sim p_{\text{data}}$, and $\mathcal{G}_t \sim q_t(\mathcal{G}_t \mid \mathcal{G}_1)$, and $\gamma$ is a weighting factor between nodes and edges.

### 4.6 PROBABILITY PATH AND PRIOR CHOICES

A natural choice for the noise prior is the uniform distribution over the simplex, $q_0(\mathbf{x}) = \text{Dir}(\mathbf{1})$. However, graphs are typically sparse, with highly imbalanced edge distributions. In discrete diffusion, Vignac et al. (2023) has shown that the empirical marginal distribution is theoretically optimal and empirically effective.

Motivated by this result, we consider a marginal-weighted Dirichlet mixture prior: $q_0^{\text{marg}}(\mathbf{x}) = \sum_{k=1}^K m_k \text{Dir}(\mathbf{x}; \mathbf{1} + \kappa e_k)$, where $\boldsymbol{m} = (m_1, \ldots, m_K)$ denotes the empirical marginal over categories, $e_k$ is the $k$th standard basis vector, and $\kappa > 0$ is a user-specified concentration parameter.

We can directly parametrize the prior as such, but we can also approximate this prior using a model trained with the probability path $q_t^\star(\mathbf{x}_t \mid \mathbf{x}_1) := \text{Dir}(\mathbf{x}_t; \mathbf{1} + \alpha_t \mathbf{x}_1)$. In that case, we have dimension-wise: $\mathbb{E}_{\mathbf{x}_1 \sim p_{\text{data}}}\left[q_t^\star(\mathbf{x}_t \mid \mathbf{x}_1)\right] = q_0^{\text{marg}}(\mathbf{x}_0) \iff \alpha_t = \kappa$. This identity does not directly extend to the $L$-dimensional joint, since in general the $L$ dimensions are not independent. Nevertheless, when $q_t^\star$ is sufficiently close to the stationary distribution $q_0^\star$, we can leverage the following approximation:

$$q_t^\star\left(\mathbf{x}_t^{(1:L)} \mid \mathbf{x}_1^{(1:L)}\right) \approx \prod_{l=1}^L q_t^\star\left(\mathbf{x}_t^{(l)} \mid \mathbf{x}_1^{(l)}\right) = q_0^{\text{marg}}(\mathbf{x}^{(1:L)}) \quad \text{with } \alpha_t = \kappa.\tag{8}$$

Operationally, we can thus initialize inference with $\mathbf{x}_0^{(1:L)} \sim q_0^{\text{marg}}$, and then run the reverse process along the path $q_t^\star$ with $\alpha_0 = \kappa$. We show in Section 5 that this initialization improves generative performance.

### 4.7 GUIDANCE

While high-quality unconditional generation is a prerequisite, conditioning on graph-level properties is necessary for many downstream tasks. In drug discovery, for example, one often seeks molecules that are both synthetically accessible and highly active against specified targets. Denoising-based models typically steer samples via either *classifier guidance* or *classifier-free guidance*; our framework supports both, because these conditioning mechanisms are direct adaptations of established techniques. Due to space limitations, we provide implementation details in Appendix B. We demonstrate the effectiveness of our approach for property-conditioned molecular generation in Section 5.

## 5 EVALUATION AND RESULTS

We evaluate our model on both molecular and synthetic graph datasets. For molecular data, we adopt two standard benchmark datasets: `QM9` and `ZINC250K`. Regarding `QM9`, we prefer the more challenging version *with explicit hydrogens* (`QM9H`), as the smaller version reaches saturation. The `QM9` dataset contains 133,885 molecules with up to 29 atoms of 5 types. The `ZINC250K` dataset contains 250,000 molecular graphs with up to 38 heavy atoms of 9 types. For generic graphs, we run experiments on the `Planar` and `Stochastic Block Model` (SBM) datasets. Both contain 200 unattributed graphs with 64 and up to 200 nodes, respectively. Visualizations of generated molecules and graphs are available in Appendix F. Our code will be released upon acceptance.

### 5.1 BASELINES

We compare against two categories of baselines: (i) published models representing different denoising approaches, and (ii) our own implementations for controlled comparisons.

Table 1: Results on Qm9H.

| | VALID (%)↑ | FCD ↓ | NSPDK↓ |
|---|---|---|---|
| TRAIN. | 98.90 | 0.062 | 0.121 |
| DIGRESS | 95.4 | | |
| DISCDIF | 22.29 | 4.246 | 41.932 |
| NM-DD | 97.97 | 0.366 | 1.149 |
| DIRIFM | 92.20 | 0.356 | 0.495 |
| UNSIDE | **98.87** | **0.152** | **0.487** |

Table 2: Results on ZINC250K.

| | VALID (%)↑ | FCD ↓ | NSPDK↓ |
|---|---|---|---|
| TRAIN. | 100.00 | 1.13 | 0.10 |
| DIGRESS | 94.99 | 3.48 | 2.1 |
| GRUM | 98.65 | 2.26 | 1.5 |
| GRABFN | 99.22 | 2.12 | 1.3 |
| SID | 99.50 | 2.06 | 2.01 |
| DISCDIF | 74.17 | 4.78 | 4.08 |
| NM-DD | 99.92 | 2.65 | 3.48 |
| DIRIFM | 97.32 | 2.79 | 1.93 |
| UNSIDE | **99.98** | **1.79** | **1.00** |

Table 3: Results on Planar.
Table 4: Results on SBM.

| | VALID (%)↑ | DEGREE ↓ | CLUST.↓ | ORBIT ↓ | SPECT. ↓ | VALID (%)↑ | DEGREE ↓ | CLUST.↓ | ORBIT ↓ | SPECT. ↓ | NOV. (%)↑ |
|---|---|---|---|---|---|---|---|---|---|---|---|
| TRAIN. | 100.0 | 0.25 | 36.6 | 0.8 | 6.6 | 93.5 | 1.57 | 50.1 | 37.0 | 4.5 | |
| DIGRESS | 45.5 | 0.71 | 45.4 | 1.31 | 8.4 | 51.0 | **1.28** | 51.5 | **39.6** | **5.0** | 100 |
| GRUM | 91.0 | **0.38** | 40.7 | 6.42 | **7.9** | 67.5 | 2.20 | **49.9** | 40.4 | 5.1 | 100 |
| SID | 91.3 | 5.9 | 163.4 | 19.1 | 7.6 | 63.5 | 11.5 | **51.4** | 123.1 | 5.9 | 100 |
| DEFOG | 99.5 | 0.5 | 50.1 | 0.6 | 7.2 | **90.0** | 0.6 | 51.7 | 55.6 | 5.4 | 90.0 |
| DISCDIF | 00.0 | 56.1 | 294.0 | 1410.0 | 85.1 | 0.0 | **1.82** | 86.4 | 125.7 | 11.2 | 100 |
| NM-DD | 98.0 | 14.1 | 363.0 | 27.2 | **6.9** | 60.5 | 4.38 | 50.9 | 52.9 | 5.7 | 100 |
| DIRIFM | 0.0 | 6.4 | 196.5 | 45.0 | 21.7 | 46.0 | 4.26 | 53.0 | 53.0 | **5.0** | 100 |
| UNSIDE | **100.0** | **0.36** | **39.9** | **0.78** | **7.1** | 78.5 | **1.74** | **49.9** | 52.1 | 5.9 | 100 |

We report results for four representative methods. DIGRESS (Vignac et al., 2023) implements standard discrete diffusion; GRUM (Jo et al., 2024) performs denoising in a continuous space; GraphBFN (GRABFN) is a Bayesian flow–network–based generative model; SID (Boget, 2025) is a non-Markovian discrete diffusion method; and DEFOG (Qin et al., 2025) is a discrete flow-matching model. In general, we reproduce the numbers reported in the original papers, except for the Zinc250K results of DIGRESS, for which we include the values provided in Jo et al. (2024), as DIGRESS does not report on this dataset. For the Planar and SBM datasets, we reran experiments for baselines that reported results from a *single* sampling run. We motivate this choice and describe our reproduction protocol in Appendix D.6. Finally, we also report metrics computed on random samples of the training set, each containing the same number of graphs as the generated samples (TRAIN.).

To isolate the effect of our sampling formulation, we train two models with identical architectures and training hyperparameters for each experiment: a discrete denoiser and a simplex-based denoiser. The only notable difference is that the discrete denoiser includes the extra input features used by DIGRESS. For each denoiser, we evaluate two sampling procedures. For the discrete denoiser, we use standard discrete diffusion (DISCDIFF, as in DIGRESS) and its non-Markovian variant (NM-DISCDIFF, as in SID). On the simplex, we implement the Dirichlet flow-matching sampler of Stark et al. (2024) (DIRIFM), as well as our unrestrained simplex denoising (UNSIDE) sampler. Experimental and implementation details are provided in Appendix D.5.

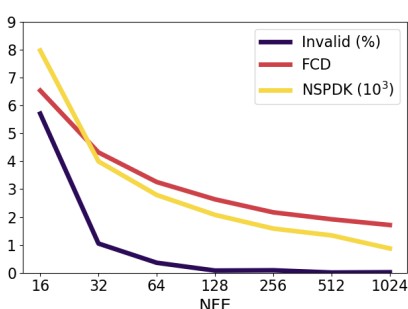

Figure 3: Evolution of metrics on ZINC250K as a function of the number of function evaluations (NFE).

## 5.2 RESULTS

We report means over five independent sampling runs. The evaluation protocol, additional metrics, and standard deviations (std.) are provided in Appendices D.5 and E. We highlight in bold all results that fall within the training-set error margin. If none do, we bold the best result.

**Molecule Generation** We evaluate generative performance by measuring distributional similarity in chemical space (FCD), subgraph structures (NSPDK, reported $\times 10^3$), and chemical validity

without correction. Our UNSIDE model surpasses all baselines, often by a significant margin, on almost all metrics, except for FCD on QM9H. In particular, it generates very few invalid molecules.

**Unattributed Graph Generation**   For unattributed graphs, we follow the procedure of Martinkus et al. (2022). We measure distributional similarity via Maximum Mean Discrepancy (MMD) over degree distributions, clustering coefficients, orbit counts, and spectral densities (MMDs are reported $\times 10^3$). We also report validity, i.e., planarity and statistical consistency with the true stochastic block model.

On Planar (Table 3), our model reaches a perfect validity rate, while also achieving comparable results to the best baseline on other metrics. On SBM (Table 4), with the exception of DEFOG, which exhibits mode-collapse issues, we again obtain the highest validity and achieve results on the remaining metrics that are comparable to the strongest baseline.

### 5.3 ABLATIONS AND EFFICIENCY

We study the impact of key design choices: the unrestrained denoising formulation, the number of function evaluations (NFE), the noise scheduler, the sampling efficiency, and the initialization of the sampling noise. First, DIRIFM and our unrestrained simplex denoising use the same denoiser; the only difference lies in the sampling strategy. Across all experiments and metrics, our unrestrained sampler consistently outperforms simplex flow matching, often by a large margin, demonstrating the empirical superiority of our sampling method. Second, we ablate NFE on ZINC250K. Results are shown in Figure 3 and Table 7. Our model attains state-of-the-art validity in as few as 64 steps, with competitive FCD and NSPDK. As expected, all metrics improve monotonically with larger NFE. Third, Appendix E.1 shows that our model is highly efficient at sampling: it is the fastest among all compared methods and achieves these results using fewer parameters than the strongest baselines. Fourth, Appendix E.2.2 demonstrates that Voronoi probabilities provide an effective tool for calibrating our noise scheduler, and that our model remains robust under substantial variations of the scheduling parameter. Finally, we compare two noise initializations: the uniform prior $q_0 = \mathrm{Dir}(\mathbf{1})$ versus the marginal mixture $q_0^{\mathrm{marg}}$ with $\kappa = 2$ (Section 4.6). Marginal initialization yields a small, but consistent improvement in all metrics (See Figure 5).

### 5.4 GUIDANCE

We evaluate conditional generation toward a target molecular property using the Quantitative Estimation of Drug-likeness (QED) score (as implemented in RDKit). On QM9H, we apply classifier guidance to steer samples toward prescribed standardized QED values. Concretely, we draw 1,000 target QEDs from the test set, generate 1,000 molecules conditioned on these targets, and compute the QEDs of the generated molecules with RDKit. Guidance effectiveness is quantified by the mean absolute error (MAE) between the target and generated QEDs.

Results are reported in Table 9. We observe that classifier guidance effectively steers generation toward the prescribed QED values (reducing MAE from 1.06 to 0.42) without hindering the model's ability to generate valid molecules.

## 6 CONCLUSION

We introduced *Unrestrained Simplex Denoising* (UNSIDE), a new denoising paradigm for categorical data that operates directly on the probability simplex and relaxes the Markovian constraints commonly imposed by diffusion and flow-matching methods. On molecular and synthetic graph benchmarks, UNSIDE matches or surpasses strong discrete diffusion and flow-matching baselines, reaches state-of-the-art validity with few denoising steps, and enables effective property-conditioned generation. Ablations demonstrate the benefits of the non-Markovian formulation, a favorable NFE–quality trade-off, and the utility of marginal-based initialization on the simplex. More broadly, UNSIDE represents a new class of simplex-based generative models, opening promising directions such as extending the framework to additional discrete modalities (e.g., sequences and structured tabular data), exploring alternative probability paths, and integrating hybrid discrete–continuous variables.

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

# A  PROOFS

## A.1  VORONOI PROBABILITY

Let recall the Gamma function:

$$\Gamma(a) = \int_0^\infty t^{a-1} e^{-t} dt, \quad a \in \mathbb{R}_{>0}, \tag{9}$$

the multinomial Beta function

$$\mathrm{B}(\boldsymbol{a}) = \frac{\prod_{i=1}^K \Gamma(a_i)}{\Gamma(\sum_{i=1}^K a_i)}, \tag{10}$$

the density function of the Dirichlet distribution for a point $\mathbf{x} \in \mathbb{S}_K$

$$f(\mathbf{x}) = \frac{1}{\mathrm{B}(\boldsymbol{\alpha})} \prod_{i=1}^{n+1} x_i^{\alpha_i - 1}, \quad \alpha_0 = \sum_i \alpha_i, \tag{11}$$

and the Gamma(a, b) density function:

$$f(x; a, b) = \frac{b^a}{\Gamma(a)} x^{a-1} e^{-bx}, \quad x > 0, \ a > 0, \ b > 0. \tag{12}$$

.

We show that if $\mathbf{x} \sim \mathrm{Dir}(1, \ldots, a, \ldots, 1)$, i.e. all parameters are ones except the $i$th parameter, which is $a$:

$$P_{v_i}(\mathbf{x} \mid \boldsymbol{e}_i) = \sum_{k=0}^{K-1} \frac{(-1)^k \binom{n}{k}}{(k+1)^{a-1}} \tag{13}$$

We remind that the Voronoi region of vertex $i$ is defined as:

$$\mathbb{V}_i = \{\boldsymbol{x} \in \mathbb{S}_K \mid d(\boldsymbol{x}, \boldsymbol{e}_i) \le d(\boldsymbol{x}, \boldsymbol{e}_j) \ \forall j \ne i\}, \tag{14}$$

with $d(\cdot, \cdot)$ the Euclidean distance.

We first note that:

$$\mathbf{x} \in \mathbb{V}_i \iff x_i = \max_{j \in [K]} x_j, \tag{15}$$

so that

$$P_{v_i}(\mathbf{x} \mid \boldsymbol{e}_i) = P(x_i = \max_{j \in [K]} x_j) \tag{16}$$

It is well-known that for any $b > 0$:

$$g_1 \sim \Gamma(\alpha_1, b), \ \ldots, \ g_K \sim \Gamma(\alpha_K, b) \implies [x_1, \ldots, x_K]^T \sim \mathrm{Dir}(\alpha_1, \ldots, \alpha_K), \tag{17}$$

where $x_i = \frac{g_i}{\sum_{j=1}^K g_j}$. Since the denominator is a constant, it does not change the ordering:

$$g_i = \max_{j \in [K]} g_j \iff x_i \max_{j \in [K]} x_j \tag{18}$$

In the following, we choose $b = 1$, and denote.

$$g_i \sim \mathrm{Gamma}(a, 1), g_j \sim \mathrm{Gamma}(1, 1), \ \forall \, j \ne i \tag{19}$$

From Equations 16 and 18 and independence of the $g_i \; \forall \; i \neq j$, we get:

$$P_{v_i}(\mathbf{x} \mid e_i) = P(g_i \geq g_1, \ldots, g_{i-1}, g_{i+1}, \ldots, g_K) \tag{20}$$

$$= P(g_i \geq g_j)^{(K-1)} \tag{21}$$

Let recall that the cumulative distribution function for $\mathrm{Gamma}(x; 1, 1)$ is:

$$F(x; 1, 1) = 1 - e^{-x} \tag{22}$$

Hence,

$$P(x \geq g_j)^{(K-1)} = (1 - e^{-x})^{(K-1)} \tag{23}$$

.

$$P(g_i \geq g_1, \ldots, g_{i-1}, g_{i+1}, \ldots, g_K) = \frac{1}{\Gamma(a)} \int_0^\infty t^{a-1} e^{-t} (1 - e^{-t})^{K-1} dt. \tag{24}$$

.

By the binomial theorem, we have:

$$(1 - e^{-t})^{K-1} = \sum_{k=0}^{K-1} (-1)^k \binom{K-1}{k} e^{-kt}. \tag{25}$$

Substituting the expression in Equation 24 and rearranging the terms, we get:

$$P_{v_i}(\mathbf{x} \mid e_i) = \frac{1}{\Gamma(a)} \sum_{k=0}^{K-1} (-1)^k \binom{K-1}{k} \int_0^\infty t^{a-1} e^{-(k+1)t} dt \tag{26}$$

Set $u = (k+1)t \implies t = \frac{u}{(k+1)}$:

$$= \frac{1}{\Gamma(a)} \sum_{k=0}^{K-1} (-1)^k \binom{K-1}{k} \int_0^\infty \frac{u}{(k+1)^{a-1}} e^{-u} \frac{1}{k+1} du \tag{27}$$

$$= \frac{1}{\Gamma(a)} \sum_{k=0}^{K-1} \frac{(-1)^k \binom{K-1}{k}}{(k+1)^a} \int_0^\infty u^{a-1} e^{-u} du \tag{28}$$

$$= \frac{1}{\Gamma(a)} \sum_{k=0}^{K-1} \frac{(-1)^k \binom{K-1}{k}}{(k+1)^a} \Gamma(a) \tag{29}$$

$$= \sum_{k=0}^{K-1} \frac{(-1)^k \binom{K-1}{k}}{(k+1)^a} \tag{30}$$

## A.2 DENOISING KERNEL

Given $P_{1|t}(\mathbf{x}_1 \mid \boldsymbol{x}_t^{(1:L)})$ and Assumption 4.1, the one-step denoising kernel satisfies

$$p_{t+dt}(\mathbf{x} \mid \boldsymbol{x}_t^{(1:L)}) = \sum_{\mathbf{x}_1} q_{t+dt}(\mathbf{x} \mid \mathbf{x}_1) P_{1|t}(\mathbf{x}_1 \mid \boldsymbol{x}_t^{(1:L)}) \tag{31}$$

$$= \mathbb{E}_{\mathbf{x}_1 \sim P_{1|t}(\cdot \mid \boldsymbol{x}_t^{(1:L)})} [q_{t+dt}(\mathbf{x} \mid \mathbf{x}_1)]. \tag{32}$$

By the low of total probabilities (first equality), and Assumption 4.1 (second equality), we have:

$$p_{t+dt}(\mathbf{x} \mid \boldsymbol{x}_t^{(1:L)}) = \sum_{\mathbf{x}_1} p_{t+dt}(\mathbf{x} \mid, \boldsymbol{x}_t^{(1:L)}, \mathbf{x}_1) P_{1|t}(\mathbf{x}_1 \mid \boldsymbol{x}_t^{(1:L)}) \tag{33}$$

$$= \sum_{\mathbf{x}_1} q_{t+dt}(\mathbf{x} \mid, \mathbf{x}_1) P_{1|t}(\mathbf{x}_1 \mid \boldsymbol{x}_t^{(1:L)}) \tag{34}$$

## A.3 CONVERGENCE

**Proposition A.1 (Convergence to the stationary distribution)** *Let assume that $P_{1|t}(\mathbf{x}_1 \mid \boldsymbol{x}_t)$ and $q(\boldsymbol{x}_t \mid \mathbf{x}_1)$ have full support on $\mathcal{S}_K^L$ and $\mathcal{S}_K^L$, respectively. Further assume that the conditional independence holds: $q_t(\boldsymbol{x}_t' \mid \boldsymbol{x}_t, \mathbf{x}_1) = q_t(\boldsymbol{x}_t' \mid \mathbf{x}_1)$.*

*Then the Markov kernel*

$$p_{t'|t}(\boldsymbol{x}_t' \mid \boldsymbol{x}_t) = \sum_{\mathbf{x}_1 \in \mathcal{S}^L} P_{1|t}(\mathbf{x}_1 \mid \boldsymbol{x}_t) q(\boldsymbol{x}_t' \mid \mathbf{x}_1). \tag{35}$$

*converge to the stationary distribution*

$$\pi(\mathbf{x}_t) = \sum_{\mathbf{x}_1 \in \mathcal{S}^L} p(\mathbf{x}_1) q(\mathbf{x}_t \mid \mathbf{x}_1), \tag{36}$$

Assume $P_{1|t}(\mathbf{x}_1 \mid \boldsymbol{x}_t)$ has full support on $\mathcal{S}^L$, and $q_t(\boldsymbol{x}_t \mid \mathbf{x}_1)$ has full support on $\mathbb{S}_K^L$. Assume also the conditional independence $q(\boldsymbol{x}_t' \mid \boldsymbol{x}_t, \mathbf{x}_1) = q(\boldsymbol{x}_t' \mid \mathbf{x}_1)$.

Then

$$p_{t'|t}(\boldsymbol{x}_t' \mid \boldsymbol{x}_t) = \sum_{\mathbf{x}_1 \in \mathcal{S}^L} P_{1|t}(\mathbf{x}_1 \mid \boldsymbol{x}_t) q(\boldsymbol{x}_t' \mid \mathbf{x}_1) \tag{37}$$

defines a valid Markov kernel on $\mathbb{S}_K^L$.

**Stationarity**  A distribution $\pi$ on $\mathbb{S}_K^L$ is stationary for $p_{t'|t}$ if

$$\pi(\boldsymbol{x}_t') = \int_{\mathbb{S}_K^L} p_{t'|t}(\boldsymbol{x}_t' \mid \boldsymbol{x}_t) \pi(\boldsymbol{x}_t) d\boldsymbol{x}_t. \tag{38}$$

Expanding the kernel and interchanging sum and integral,

$$\pi(\boldsymbol{x}_t') = \int_{\mathbb{S}_K^L} \pi(\boldsymbol{x}_t) \sum_{\mathbf{x}_1 \in \mathcal{S}^L} P_{1|t}(\mathbf{x}_1 \mid \boldsymbol{x}_t) q(\boldsymbol{x}_t' \mid \mathbf{x}_1) d\boldsymbol{x}_t \tag{39}$$

$$= \sum_{\mathbf{x}_1 \in \mathcal{S}^L} \int_{\mathbb{S}_K^L} \pi(\boldsymbol{x}_t) P_{1|t}(\mathbf{x}_1 \mid \boldsymbol{x}_t) d\boldsymbol{x}_t \, q(\boldsymbol{x}_t' \mid \mathbf{x}_1). \tag{40}$$

By the law of total probability,

$$\int_{\mathbb{S}_K^L} \pi(\boldsymbol{x}_t) P_{1|t}(\mathbf{x}_1 \mid \boldsymbol{x}_t) d\boldsymbol{x}_t = p(\mathbf{x}_1), \tag{41}$$

the marginal distribution of $\mathbf{x}_1$. Hence

$$\pi(\boldsymbol{x}_t') = \sum_{\mathbf{x}_1 \in \mathcal{S}^L} p(\mathbf{x}_1) q(\boldsymbol{x}_t' \mid \mathbf{x}_1). \tag{42}$$

Thus the stationary distribution of $p_{t'|t}$ is the marginal

$$\pi = \sum_{\mathbf{x}_1 \in \mathcal{S}^L} p(\mathbf{x}_1) q(\cdot \mid \mathbf{x}_1). \tag{43}$$

**Convergence.** Since both $P_{1|t}$ and $q(\cdot \mid \mathbf{x}_1)$ have full support, the kernel $p_{t'|t}$ has strictly positive density on the simplex. This ensures irreducibility and aperiodicity. Then, by the Fundamental Theorem of Markov Chains (Biswas, 2022), the distribution of $\mathbf{x}_t$ converges to $\pi$ for any initialization.

## B GUIDANCE

Classifier and classifier-free guidance are two standard mechanisms for steering generation toward desired properties. Classifier guidance augments an *unconditional* generative model with the gradient of a separately trained classifier (or regressor) for the target property. Classifier-free guidance does not require an auxiliary classifier, but it does require both a *conditional* and an *unconditional* generative model, typically obtained by jointly training with masked/unmasked conditioning vectors.

Below we first present the high-level ideas, then provide derivations.

### B.1 KEY IDEAS

Let $y$ denote the target property and let $p_y(y \mid \mathbf{x})$ be a discriminative model (classifier/regressor) of $y$ given $\mathbf{x}$. Let $p_{t+dt|t}(\mathbf{x}_{t+dt} \mid \mathbf{x}_t)$ denote the unconditional reverse transition at time $t$. We modulate the strength of conditioning with a scale $\omega > 0$ via

$$p_{t+dt}^{\omega}(\mathbf{x}|\mathbf{x}_t, y) \propto p_y(y|\mathbf{x}_{t+dt})^{\omega} p_{t+dt}(\mathbf{x}_{t+dt}|\mathbf{x}_t) \tag{44}$$

.

**Classifier guidance.** Here $p_y(y \mid \mathbf{x})$ is provided by a classifier trained for all noise levels. During sampling, one combines a reverse step with an update in the direction of the classifier gradient, $\nabla_{\mathbf{x}_t} \log p_{y|x}^{\phi}(y \mid \mathbf{x}_t)$.

**Classifier-free guidance.** Within this approach, no external classifier is needed; instead, one uses a conditional reverse model $p_{t+dt|t,y}(\mathbf{x}_{t+dt} \mid \mathbf{x}_t, y)$ and an unconditional one $p_{t+dt|t}(\mathbf{x}_{t+dt} \mid \mathbf{x}_t)$, typically produced by a single network via masking. A standard identity yields the log-linear interpolation

$$\log p_{t+dt|t,y}^{\omega}(\mathbf{x}_{t+dt}|\mathbf{x}_t, y) \propto \omega \log \left( p_{t+dt|t,y}(\mathbf{x}_{t+dt}|\mathbf{x}_t, y) \right) + (1-\omega) \log \left( p_{t+dt|t}(\mathbf{x}_{t+dt}|\mathbf{x}_t) \right). \tag{45}$$

### B.2 DERIVATIONS

Assume the corruption (noising) process is independent of the conditioning variable $y$: $q_t(\mathbf{x}_t \mid \mathbf{x}_{t+dt}, y) = q_t(\mathbf{x}_t \mid \mathbf{x}_{t+dt})$. By Bayes' rule and this independence,

$$p_y(y|\mathbf{x}_t, \mathbf{x}_{t+dt}) = \frac{q_{t|y,t+dt}(\mathbf{x}_t|\mathbf{x}_{t+dt}, y) p_y(y|\mathbf{x}_{t+dt})}{q_{t|y,t+dt}(\mathbf{x}_t|\mathbf{x}_{t+dt})} \tag{46}$$

$$= \frac{q_{t|y,t+dt}(\mathbf{x}_t|\mathbf{x}_{t+dt}) p_y(y|\mathbf{x}_{t+dt})}{q_{t|y,t+dt}(\mathbf{x}_t|\mathbf{x}_{t+dt})} \tag{47}$$

$$= p_y(y|\mathbf{x}_{t+dt}) \tag{48}$$

Consequently,

$$p_{t+dt|t,y}(\mathbf{x}_{t+dt}|\mathbf{x}_t, y) = \frac{p_y(y|\mathbf{x}_{t+dt}, \mathbf{x}_t) p_{t+dt|t}(\mathbf{x}_{t+dt}|\mathbf{x}_t)}{p_y(y|\mathbf{x}_t)} \tag{49}$$

$$\propto p_y(y|\mathbf{x}_{t+dt}) p_{t+dt|t}(\mathbf{x}_{t+dt}|\mathbf{x}_t). \tag{50}$$

Introducing $\omega > 0$ sharpens the guidance:

$$p_{t+dt|t,y}^{\omega}(\mathbf{x}_{t+dt}|\mathbf{x}_t, y) \propto p_y(\mathbf{x}_{t+dt}|\mathbf{x}_{t+dt})^{\omega} p_{t+dt|t}(\mathbf{x}_{t+dt}|\mathbf{x}_t) \tag{51}$$

### B.2.1 CLASSIFIER FREE GUIDANCE

Applying bayes' rule again on the last Equation, we obtain:

$$p_{t+dt|t,y}^{\omega}(\mathbf{x}_{t+dt}|\mathbf{x}_t, y) \propto p_y(y|\mathbf{x}_{t+dt})^{\omega} p_{t+dt|t}(\mathbf{x}_{t+dt}|\mathbf{x}_t) \tag{52}$$

$$= \left( \frac{p(\mathbf{x}_{t+dt}|y, \mathbf{x}_t) p(y|\mathbf{x}_t)}{p(\mathbf{x}_{t+dt}|\mathbf{x}_t)} \right)^{\omega} p_{t+dt|t}(\mathbf{x}_{t+dt}|\mathbf{x}_t) \tag{53}$$

$$= p(\mathbf{x}_{t+dt}|y, \mathbf{x}_t)^{\omega} \frac{p(\mathbf{x}_{t+dt}|\mathbf{x}_t)}{p_{t+dt|t}(\mathbf{x}_{t+dt}|\mathbf{x}_t)^{\omega}} p(y|\mathbf{x}_t)^{\omega} \tag{54}$$

By taking the logarithm and pushing the last term on the right into the constant we get:

$$\log p_{t+dt|t,y}^{\omega}(\mathbf{x}_{t+dt}|\mathbf{x}_t, y) \propto \omega \log \left( p_{t+dt|t,y}(\mathbf{x}_{t+dt}|\mathbf{x}_t, y) \right) + (1-\omega) \log \left( p_{t+dt|t}(\mathbf{x}_{t+dt}|\mathbf{x}_t) \right). \tag{55}$$

### B.2.2 CLASSIFIER GUIDANCE

For classifier guidance, we first apply Bayes' rules two times to Equation 51

$$p_{t+dt|t,y}^{\omega}(\mathbf{x}_{t+dt}|\mathbf{x}_t, y) \propto p_y(y|\mathbf{x}_{t+dt})^{\omega} p_{t+dt|t}(\mathbf{x}_{t+dt}|\mathbf{x}_t) \tag{56}$$

$$= \left( \frac{p(\mathbf{x}_{t+dt}|y, \mathbf{x}_t) p(y|\mathbf{x}_t)}{p(\mathbf{x}_{t+dt}|\mathbf{x}_t)} \right)^{\omega} p_{t+dt|t}(\mathbf{x}_{t+dt}|\mathbf{x}_t) \tag{57}$$

$$= p_{t+dt|t}(\mathbf{x}_{t+dt}|y, \mathbf{x}_t)^{\omega} \frac{p(\mathbf{x}_{t+dt}|\mathbf{x}_t)}{p_{t+dt|t}(\mathbf{x}_{t+dt}|\mathbf{x}_t)^{\omega}} p(y|\mathbf{x}_t)^{\omega} \tag{58}$$

$$= p(\mathbf{x}_{t+dt}|\mathbf{x}_t) p(y|\mathbf{x}_t)^{\omega} \tag{59}$$

We then use Taylor approximation to evaluate $p(y|\mathbf{x}_t)$:

$$\log \tilde{p}_{y|t+dt}(y|\mathbf{x}_{t+dt}) = \mathbf{x}_{t+dt}^T + dt \nabla_z \log p_{y|z}(y|z)|_{z=\mathbf{x}_t} + C \tag{60}$$

$$= \sum_{i=1}^{L} z^T \nabla_z \log p_{y|z}(y|z)|_{z=\mathbf{x}_t} + C \tag{61}$$

$$= \sum_{i=1}^{L} \frac{\partial}{\partial z_j^{(i)}} \log p_{y|z}(y|\mathbf{x}_t)|_{z=\mathbf{x}_t} + C, \tag{62}$$

which gives:

$$\log p_{t+dt|t,y}^{\omega}(\mathbf{x}_{t+dt}|\mathbf{x}_t, y) \propto p_{t+dt|t}(\mathbf{x}_{t+dt}|\mathbf{x}_t) + \omega \nabla_{\mathbf{x}_t} \log p_{y|z}(y|\mathbf{x}_t), \tag{63}$$

This shows that classifier guidance amounts to a gradient-ascent adjustment of the unconditional reverse kernel toward higher discriminative likelihood of the target property.

## C COMPLEMENTARY TEXTS

### C.1 EXTENDED RELATED WORK ON GRAPH GENERATION

A central challenge in generative graph modeling follows from the many $n!$ different ways to represent graphs due to node permutation. This has motivated two dominant families of methods: *sequential* (autoregressive) approaches generating graphs step by step, adding nodes, edges, or higher-order motifs according to a fixed or learned ordering (You et al., 2018; Shi et al., 2020; Luo et al., 2021; Liao et al., 2019; Goyal et al., 2020; Kong et al., 2023; Zhao et al., 2024), and *permutation-equivariant* models that enforce equivariance by design. Permutation-equivariant architectures have been instantiated within several generative frameworks, including variational autoencoders (Simonovsky & Komodakis, 2018), generative adversarial networks (De Cao & Kipf,

2018; Krawczuk et al., 2021; Martinkus et al., 2022), normalizing flows (Madhawa et al., 2019; Zang & Wang, 2020; Liu et al., 2019), and vector-quantized autoencoders (Boget et al., 2024; Nguyen et al., 2024).

Orthogonal to this point, a recent line of work targets scalability. A central limitation of standard models is their reliance on dense representations and all-pairs computations, which impedes scaling to large graphs. To address this, prior work has proposed scalable denoising architectures and sampling strategies (Qin et al., 2024; Chen et al., 2023; Karami, 2024; Bergmeister et al., 2024; Boget et al., 2025). Notably, Qin et al. (2024) and Boget et al. (2025) introduce methods that enable permutation-equivariant models to scale to large graphs. These techniques are complementary to our approach and could be used to scale our unrestrained simplex denoising framework; we leave this exploration to future work.

## C.2 DENOISER EXPRESSIVITY IN DISCRETE AND CONTINUOUS SETTINGS

Graph neural networks (GNNs) exhibit limited expressivity due to inherent graph symmetries (Morris et al., 2019). These limitations, however, disappear once each node is equipped with a unique identifier. In particular, Loukas (2020) demonstrates that sufficiently deep message-passing architectures become universal when nodes can be uniquely distinguished. Continuous noise naturally provides such distinguishing information, effectively acting as a unique identifier for each node. Consequently, injecting continuous noise enhances the expressive power of GNN-based denoisers.

In practice, discrete denoisers often require additional hand-crafted features to overcome symmetry-induced limitations. Common examples include spectral embeddings derived from Laplacian eigenvectors (Vignac et al., 2023) and relative random-walk probability features (Qin et al., 2025). These computations must be performed before each training and sampling pass, introducing a non-negligible computational overhead.

A typical criticism of continuous models is that adding continuous noise to edge attributes may collapse the graph into a fully connected weighted structure, ostensibly erasing the discrete adjacency information. However, the graph operations underlying the aforementioned extra features—such as spectral decompositions and random-walk statistics—remain well-defined on weighted graphs. Thus, continuous perturbations do not destroy the structure required for these computations. However, since continuous noise serves as an intrinsic node identifier, it eliminates the need for auxiliary feature computations.

## C.3 COMPARISON WITH SIMPLE ITERATIVE DENOISING

Our `Unside` framework can be viewed as a continuous analogue of *Simple Iterative Denoising* (SID). Below we outline the key similarities and differences.

**Noising.** In SID, the corruption process follows discrete diffusion for categorical data (Austin et al., 2021), where intermediate noisy distributions linearly interpolate between the data distribution and a stationary prior:

$$q_{t|1}(\mathbf{x} \mid \mathbf{x}_1) = \alpha_t \delta_{\mathbf{x}_1}(\mathbf{x}) + (1 - \alpha_t)q_0(\mathbf{x}). \tag{64}$$

As discussed in Section 3, this interpolation is not suitable in the continuous setting on the simplex. Accordingly, `Unside` adopts an explicit probability path within the simplex.

**Independence Assumption.** Both SID and `Unside` rely on the same conditional-independence assumption (Assumption 4.1), which is central to the formulation.

**Denoising.** The denoising rules in `Unside` follow directly from the conditional-independence assumption and are closely related in spirit to SID, even if Equation 5 does not appear in SID explicitly. A notable distinction is that our method couples a *continuous* kernel $q_t$ on the simplex with a *discrete* posterior $P_{1|t}$, an original hybrid construction that is part of our contribution.

**Convergence.** Proposition 4.3 and its proof are original to this work. SID provides no analogous convergence result. We regard this result as a core theoretical contribution.

**Advantage over Diffusion and Flow Matching**   Although there are conceptual similarities at a high level, including the connection to compounding denoising errors, the underlying intuition differs substantially between the continuous and discrete settings. In particular, the observation that $\mathbf{x}_{t+dt}$ remains close to $\mathbf{x}_t$, and the reasoning that follows from this, holds only in the continuous case and does not translate directly to discrete models.

**Parametrization and learning.**   Both approaches employ standard parameterizations and training objectives common to discrete diffusion and flow-matching methods.

**Probability paths and priors.**   Our choices and analysis of probability paths and priors are specific to continuous simplex noise and have no direct counterpart in SID.

**Guidance.**   As explained in the main text, guidance in `Unside` is obtained via direct adaptations of standard diffusion techniques (classifier and classifier-free guidance). SID does not include a guidance mechanism.

Beyond this element-wise comparison, our primary contribution is to integrate these components into a simple and efficient framework for graph generation that achieves state-of-the-art performance across multiple datasets.

## C.4   USE OF LLMs

We employed large language models (LLMs) only for editorial purpose, identifying typographical errors, and formatting tables. No scientific content, code, analyses, or results were generated by LLMs.

# D   TECHNICAL REPORT

## D.1   GNNs ARCHITECTURE

Our model is built on Simple Iterative Denoising (Boget, 2025). We use the same architecture and reproduce the architecture description.

The denoisers are Graph Neural Networks, inspired by the general, powerful, scalable (GPS) graph Transformer.

A single layer is described as:

$$\tilde{\boldsymbol{X}}^{(l)}, \tilde{\boldsymbol{E}}^{(l)} = \text{MPNN}(\boldsymbol{X}^{(l)}, \boldsymbol{E}^{(l)}), \tag{65}$$

$$\boldsymbol{X}^{(l+1)} = \text{MultiheadAttention}(\tilde{\boldsymbol{X}}^{(l)} + \boldsymbol{X}^{(l)}) + \tilde{\boldsymbol{X}}^{(l)} \tag{66}$$

$$\boldsymbol{E}^{(l+1)} = \tilde{\boldsymbol{E}}^{(l)} + \boldsymbol{E}^{(l)} \tag{67}$$

where, $\boldsymbol{X}^{(l)}$ and $\boldsymbol{E}^{(l)}$ are the node and edge hidden representations after the $l^{\text{th}}$ layer. The *Multihead Attention* layer is the classical multi-head attention layer from Vaswani et al. (2017), and MPNN is a Message-Passing Neural Network layer described hereafter.

The MPNN operates on each node and edge representations as follow:

$$\boldsymbol{h}_{i,j}^l = \text{ReLU}(\boldsymbol{W}_{src}^l \boldsymbol{x}_i^l + \boldsymbol{W}_{trg}^l \boldsymbol{x}_j^l + \boldsymbol{W}_{edge}^l \boldsymbol{e}_{i,j}^l) \tag{68}$$

$$\boldsymbol{e}_{i,j}^{l+1} = \text{LayerNorm}(f_{\text{edge}}(\boldsymbol{h}_{i,j}^l) \tag{69}$$

$$\boldsymbol{x}_i^{l+1} = \text{LayerNorm}\left( \boldsymbol{x}_i^l + \sum_{j \in \mathcal{N}(i)} f_{\text{node}}(\boldsymbol{h}_{i,j}^l) \right), \tag{70}$$

with $\boldsymbol{W}_{src}^l$, $\boldsymbol{W}_{trg}^l$, and $\boldsymbol{W}_{edge}^l$ denoting trainable weight matrices, and $f$node and $f_{\text{edge}}$ being small neural networks.

The node hidden states $\boldsymbol{x}_i$ and the outputs of $f_{\text{node}}$ are of dimension $d_h$, a tunable hyperparameter (see Table 5). The edge hidden states $\boldsymbol{e}_{i,j}$, intermediate messages $\boldsymbol{h}_{i,j}^l$, and the outputs of $f_{\text{edge}}$ have dimension $d_h/4$.

**Inputs and Outputs**   In the input, we concatenate the node attributes, extra features, and time step as node features, copying graph-level information (e.g., time step or graph size) to each node. The node and edge input vectors are then projected to their respective hidden dimensions, $d_h$ for nodes and $d_h/4$ for edges.

Similarly, the outputs of the final layer are projected to their respective dimensions, $d_x$ for nodes and $d_e$ for edges (or to a scalar in the case of the *Critic*). To enforce edge symmetry, we compute $\boldsymbol{e}_{i,j} = \frac{\boldsymbol{e}_{i,j} + \boldsymbol{e}_{j,i}}{2}$. Finally, we ensure the outputs can be interpreted as probabilities by applying either a softmax or sigmoid function, as appropriate.

## D.2   PROBABILITY PATH AND SCHEDULERS

For all experiments, we use a probability path parametrize via the Dirichlet distribution: $\text{Dir}\big(\mathbf{x}_t \mathbf{1} + \alpha_t \mathbf{x}_1\big)$, where $\alpha_t = -a \log(1 - t)$. The hyperparameter $a$ defines the noising dynamic (see Figure 2). We use $a = 3$ in all our experiments, except on SBM where we use 2.

## D.3   HYPERPARAMETERS

Table 5: Hyperparameters

|  | Planar | SBM | Qm9H | Zinc250K |
|---|---|---|---|---|
| GNN layers | 8 | 8 | 8 | 8 |
| Hidden layers in MLPs | 2 | 2 | 2 | 2 |
| Node representation size | 256 | 256 | 256 | 256 |
| Edge representation size | 64 | 128 | 128 | 128 |
| Diffusion steps | 128 | 512 | 1024 | 1024 |
| Learning rate | 0.0002 | 0.0005 | 0.0005 | 0.0005 |
| Optimizer | AdamW | AdamW | AdamW | AdamW |

For discrete diffusion and SID, we use the cosine scheduler (Nichol & Dhariwal, 2021) and the marginal noise distribution.

## D.4   EXTRA FEATURES

For Discrete Diffusion (Markovian and non-Markovian, i.e. DiscDif and NM-DD) we follow a common practice (Vignac et al., 2023; Qin et al., 2025; Boget et al., 2025), and enhance the discrete graph representation with synthetic extra node features.

For our our Unside and for Dirichlet flowmatching, we only add the graph size.

We use the following extra features: eigen features (ZINC250k, Planar, SBM), graph size, molecular features (ZINC250k), and cycle information (Planar), and the Relative Random Walk Probabilities (RRWP, Qm9 and Qm9H). All these features are concatenated to the input node attributes, and edge edge attributes for the RRWPs.

**Spectral features**   We use the eigenvectors associated with the $k$ lowest eigenvalues of the graph Laplacian. Additionally, we concatenate the corresponding $k$ lowest eigenvalues to each node.

**Graph size encoding**   The graph size is encoded as the ratio between the size of the current graph and the largest graph in the dataset, $n/n_{\max}$. This value is concatenated to all nodes in the graph.

**Molecular features**   For molecular datasets, we use the charge and valency of each atom as additional features.

**Cycles**    Following Vignac et al. (2023), we count the number of cycles of size 3, 4, and 5 that each node is part of, and use these counts as features.

**Relative Random Walk Probabilities**    . It is the probabilities of reaching a target node from from a node in a given number of steps. We typically compute it for all number of steps between 1 and $k$.

## D.5    EVALUATION

### D.5.1    MOLECULE GENERATION

For the molecular graph datasets `QM9` and `Zinc250K`, we adopt the evaluation procedure followed by Jo et al. (2024), from which we took the baseline model results, and which was originally established in Jo et al. (2022). We assess performance using three standard metrics: (i) Fréchet ChemNet Distance (FCD) (Preuer et al., 2018), which measures similarity in chemical feature space; (ii) Neighborhood Subgraph Pairwise Distance Kernel (NSPDK) (Costa & Grave, 2010), which evaluates graph-structural similarity; and (iii) validity, the fraction of chemically valid molecules without any post hoc correction or resampling. We report means over five sampling runs, each generating 10,000 molecules. For completeness, we also provide standard deviations, as well as uniqueness and novelty statistics, in Appendix E.

**QM9H**    For `QM9H`, in general we follow the above procedure used for `QM9`. For evaluation, we rely on SMILES with explicit hydrogen atoms to compute validity, uniqueness, and FCD. Under this evaluation procedure, we observe that a small fraction of molecules in the dataset are invalid. We use the kekulized version of the dataset (i.e., with three possible bond types: single, double, and triple). We note that some concurrent implementations use a variant of the dataset with explicit aromatic bonds; in such cases, evaluation metrics, in particular validity and FCD are not directly comparable to ours.

### D.5.2    UNATTRIBUTED GRAPH GENERATION

For unattributed graphs, we follow the procedure of Martinkus et al. (2022): an 80/20 train–test split with 20% of the training set used for validation We measure distributional similarity via Maximum Mean Discrepancy (MMD) over degree distributions, clustering coefficients, orbit counts, and spectral densities. We measure distributional similarity via Maximum Mean Discrepancy (MMD) over degree distributions, clustering coefficients, orbit counts, and spectral densities. We additionally report validity. For the `Planar` dataset, a valid graph must be planar and connected. For the `Stochastic Block Model` (SBM) dataset, validity indicates consistency with the data-generating block model (intra-community edge density 0.3, inter-community edge density 0.005). We omit uniqueness because all models reach $100\%$ on both datasets, and we report novelty only for `SBM` (novelty is $100\%$ on `Planar`). Uniqueness is the fraction of distinct graphs among generated samples; novelty is the fraction of unique graphs not present in the training set. We report means over five runs, each generating 40 graphs.

## D.6    BASELINES

As explained in the main text, we report baseline results as presented in the corresponding original papers, except for GRUM and DiGRESS on `Planar` and `SBM`, for which we reran the experiments using the official repositories. Below, we explain the motivation for this choice and detail the procedure we followed, as we were unable to reproduce the reported results.

### D.6.1    MOTIVATION FOR MULTIPLE SAMPLING RUNS

On unattributed graph datasets such as `Planar` and `SBM`, the test sets are relatively small, and sampling-based metrics exhibit high variance across runs. Moreover, evaluating many models, checkpoints, or seeds increases the risk of overfitting to the test set. Multiple sampling runs are therefore strongly recommended, and the corresponding variability should be assessed via standard deviations.

Several baselines report results from a *single* sampling run, including GRUM, DIGRESS, and GRAPHBFN. These reported results even outperform samples drawn directly from the training set, indicating clear overfitting. For this reason, we reran the experiments for GRUM and DIGRESS. Unfortunately, the official GRAPHBFN repository is currently empty, preventing us from reproducing that baseline.

### D.6.2 PROCEDURE FOR RESULT REPRODUCTION

We were unable to reproduce the published results for either GRUM or DIGRESS. Below we describe the exact procedure we followed.

**GRUM.** We used the pretrained models available in the official repository: `https://github.com/harryjo97/GruM/tree/master/GruM_2D`. We performed five sampling runs, varying only the sampling seed, and computed metrics using the code provided by the authors. We note that the default seed yields substantially better performance than the others.

**DIGRESS.** Since no pretrained models are currently available, we retrained DIGRESS using the official repository (`https://github.com/cvignac/DiGress/`) and the configurations specified in the corresponding YAML files. As specified, we evaluated the model every 400 epochs and computed metrics against the validation set. We increased the sampling size during training to 40 samples for stability. We trained for 100,000 epochs on `Planar` and 20,000 epochs on `SBM`, selecting the checkpoint with the highest validity for final evaluation. Note, we corrected a minor inconsistency in `src/datasets/spectre_dataset.py`, where preprocessed graphs were added twice to their respective sets (lines 93 and 100).

## E ADDITIONAL RESULTS

In this Section, we provide the following additional results:

- Model sizes and Sampling Times.
- Ablation:
  - NFE.
  - Noise Schedule.
  - Comparison between Priors.
- Guidance.
- Results with Standard Deviation

### E.1 MODEL SIZES AND SAMPLING TIMES

We report the number of parameters for each model as well as the wall-clock time required to generate 100 sampling iterations for 40 `Planar` graphs and 40 `SBM` graphs. All experiments were conducted on a single server with 1 GPU (NVIDIA GeForce RTX 3090) and 128 CPU cores.

Our models are the most computationally efficient. Since Dirichlet Flow Matching (DIRIFM) and our UNSIDE share the same denoiser architecture, differences in sampling speed arise solely from the sampling procedure itself. Likewise, our implementation of SID (NN-DD) uses the same architecture as UNSIDE; the additional overhead in NN-DD is primarily due to the computation of extra structural features required by the discrete denoiser.

Table 6: Model Size and Sampling Time.

|  | PLANAR | | SBM | | QM9H | ZINC250K |
|  | TIME | PARAMS | TIME | PARAMS | PARAMS | PARAMS |
| --- | --- | --- | --- | --- | --- | --- |
| DIGRESS | $16.9 \pm 0.0$ | 8.9M | $96.9 \pm 0.1$ | 7.1M | 3.6M | 8.2M |
| GRUM | $29.3 \pm 0.0$ | 7.1M | $84.2 \pm 0.0$ | 7.1M | - | 8.2M |
| NN-DD | $9.9 \pm 0.4$ | 5.4M | $84.1 \pm 0.3$ | 6.2M | 6.2M | 6.2M |
| DIRIFM | $9.4 \pm 0.1$ | 5.4M | $89.0 \pm 0.3$ | 6.2M | 6.2M | 6.2M |
| UNSIDE | $6.3 \pm 0.0$ | 5.4M | $72.6 \pm 0.5$ | 6.2M | 6.2M | 6.2M |

NB. We assume that GRUM uses the same architecture for its implementation of DiGress on `Zinc250K` as for its own model.

## E.2 ABLATION

We conduct all ablation studies on `Zinc250K`, as it is by far the largest dataset and provides the largest test set, yielding more precise and robust evaluation metrics. This dataset is therefore the most sensitive to small effects.

### E.2.1 NFE

When ablating the number of function evaluations (NFE), we observe that our model achieves state-of-the-art validity within just 64 denoising steps, and reaches state-of-the-art performance on the remaining metrics by 256 steps. This confirms the strong efficiency and overall effectiveness of our approach.

Table 7: Effect of NFE on generation: `Zinc250K` results.

| NFE | INVALID (%) | FCD | NSPDK ($10^3$) |
|-----|-------------|------|----------------|
| 16 | 5.71 | 6.54 | 7.97 |
| 32 | 1.05 | 4.32 | 4.00 |
| 64 | 0.36 | 3.26 | 2.79 |
| 128 | 0.08 | 2.63 | 2.08 |
| 256 | 0.09 | 2.17 | 1.59 |
| 512 | 0.01 | 1.92 | 1.34 |
| 1024 | 0.02 | 1.71 | 0.87 |

### E.2.2 NOISE SCHEDULE

We assess the effect of the noise scheduler $\alpha_t$, which parametrizes our probability paths $\mathrm{Dir}\big(\mathbf{x}_t \mathbf{1} + \alpha_t \mathbf{x}_1\big)$, with $\alpha_t = -a \log(1 - t)$ Concretely, we evaluate our model across a range of values for the hyperparameter $a$. We find that our method is remarkably robust to variations in $a$.

Figure 4 illustrates the corresponding Voronoi probabilities for each tested scheduler. Since the Voronoi probability depends on the number of categories $K$, we report it separately for node attributes $K = 9$ and edge attributes $K = 2$.

From Figure 4, we observe that the scheduler with low values of $a$ (in particular with $a = 1$) induces a probability path that becomes too sharp near the end of the trajectory (i.e., as $t \to 1$), making the denoising task overly difficult at the final steps. Conversely, for large values of $a$, the probability path becomes too flat toward the end of the process, leaving the denoiser with little useful signal to correct.

The results in Table 8 align with these expectations. In particular, when $a = 1$, model performance degrades substantially. For large values of $a$, performance also decreases but only marginally. Overall, the model appears highly robust to variations in $a$, with all tested values yielding significant improvements over the baselines.

We draw two main conclusions from this experiment: (1) the Voronoi-probability analysis provides a reliable tool for calibrating the noise scheduler, and (2) our model is robust to a wide range of scheduler choices.

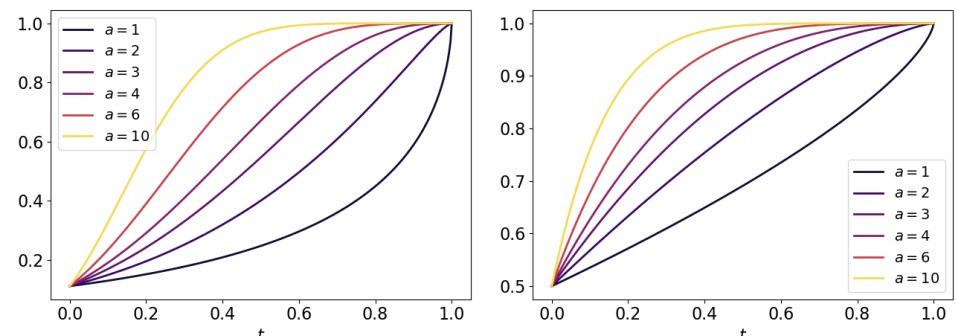

Figure 4: Voronoi probabilities for various values $a$ of the scheduler $\alpha_t = -a\log(1-t)$ with $K = 9$ (left) and $K = 2$ (right).

Table 8: Metrics for models with various values $a$ of the scheduler $\alpha_t = -a\log(1-t)$.

| A | VALID (%) | FCD | NSPDK ($10^3$) | UNIQUE | NOVEL |
|---|-----------|-----|----------------|--------|-------|
| 1 | $98.87 \pm 0.16$ | $4.13 \pm 0.08$ | $2.61 \pm 0.11$ | $99.86 \pm 0.01$ | $99.99 \pm 0.01$ |
| 2 | $99.89 \pm 0.03$ | $1.89 \pm 0.01$ | $0.98 \pm 0.06$ | $99.98 \pm 0.00$ | $99.96 \pm 0.01$ |
| 3 | $99.98 \pm 0.01$ | $1.79 \pm 0.03$ | $1.00 \pm 0.06$ | $100.00 \pm 0.00$ | $99.97 \pm 0.01$ |
| 4 | $99.94 \pm 0.02$ | $1.82 \pm 0.05$ | $1.05 \pm 0.04$ | $99.99 \pm 0.00$ | $99.98 \pm 0.01$ |
| 6 | $99.92 \pm 0.02$ | $1.66 \pm 0.03$ | $0.82 \pm 0.03$ | $99.99 \pm 0.01$ | $99.97 \pm 0.02$ |
| 10 | $99.91 \pm 0.00$ | $1.95 \pm 0.03$ | $1.25 \pm 0.07$ | $99.99 \pm 0.01$ | $99.99 \pm 0.00$ |

### E.2.3 PRIOR CHOICE

We show that the marginal weighted mixture of Dirichlet improves slightly but consistently the sampling performance across all metrics.

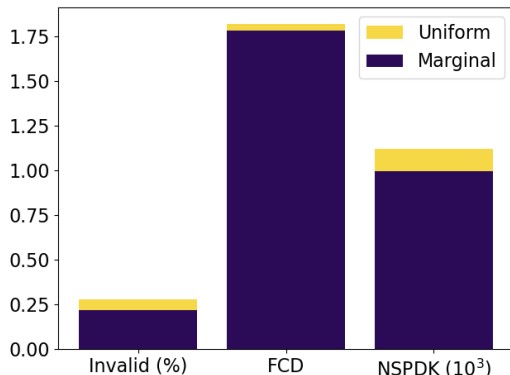

Figure 5: Comparison between uniform prior and marginal weighted mixture of Dirichlet.

### E.3 GUIDANCE

This experiment has for only purpose to show that our model support classifier guidance without affecting its ability to generate valid (molecular) graphs.

Table 9: Classifier guidance effect on QED values.

|  | VALID | MAE |
|---|---|---|
| UNCONDITIONAL | 99.6 | 1.06 |
| CLASSIFIER GUIDANCE | 99.5 | 0.42 |

## E.4 DETAILED RESULTS

Table 10: Graph generation results on `Planar`.

|  | VALID (%) | DEGREE | CLUSTER | ORBIT | SPECTRAL |
|---|---|---|---|---|---|
| TRAIN. SET | $100.0 \pm 0.0$ | $0.25 \pm 0.18$ | $36.6 \pm 3.5$ | $0.78 \pm 0.37$ | $6.62 \pm 1.34$ |
| DIGRESS | $45.5 \pm 7.0$ | $0.71 \pm 0.40$ | $45.4 \pm 13.6$ | $1.31 \pm 0.78$ | $8.40 \pm 1.16$ |
| GRUM | $91.0 \pm 2.6$ | $0.38 \pm 0.15$ | $40.5 \pm 4.8$ | $6.42 \pm 1.96$ | $7.87 \pm 1.17$ |
| DEFOG | $99.5 \pm 1.0$ | $0.5 \pm 0.2$ | $50.1 \pm 14.9$ | $0.6 \pm 0.4$ | $7.2 \pm 1.1$ |
| SID | $91.3 \pm 4.1$ | $5.93 \pm 1.26$ | $163.4 \pm 31.9$ | $19.1 \pm 4.14$ | $7.62 \pm 1.34$ |
| DISCDIF | $0.0 \pm 0.0$ | $56.1 \pm 3.70$ | $294.0 \pm 3.4$ | $1410.0 \pm 44.5$ | $85.1 \pm 1.6$ |
| NM-DD | $98.0 \pm 1.0$ | $14.1 \pm 2.58$ | $363.0 \pm 45.6$ | $27.2 \pm 5.6$ | $6.9 \pm 0.8$ |
| DIRIFM | $0.0 \pm 0.0$ | $6.44 \pm 1.42$ | $196.6 \pm 22.5$ | $45.04 \pm 5.83$ | $21.76 \pm 1.30$ |
| UNSIDE | $100.0 \pm 0.0$ | $0.36 \pm 0.24$ | $39.9 \pm 8.7$ | $0.78 \pm 0.49$ | $7.12 \pm 0.45$ |

Table 11: Graph generation results on `Stochastic Block Model`.

|  | VALID (%) | DEGREE ($\times 10^3$) | CLUSTER ($\times 10^3$) | ORBIT ($\times 10^3$) | SPECTRAL ($\times 10^3$) | NOVEL (%) |
|---|---|---|---|---|---|---|
| TRAIN. SET | $93.50 \pm 2.00$ | $1.57 \pm 0.55$ | $50.11 \pm 0.51$ | $37.0 \pm 10.7$ | $4.58 \pm 0.49$ |  |
| DIGRESS | $51.00 \pm 9.62$ | $1.28 \pm 0.48$ | $51.49 \pm 1.31$ | $39.6 \pm 7.7$ | $5.04 \pm 0.63$ | $100 \pm 0.00$ |
| GRUM | $67.54 \pm 2.98$ | $2.20 \pm 0.76$ | $49.88 \pm 0.62$ | $40.4 \pm 5.6$ | $5.06 \pm 0.72$ | $100 \pm 0.00$ |
| SID | $63.5 \pm 3.7$ | $11.5 \pm 2.7$ | $51.4 \pm 1.5$ | $123. \pm 5.4$ | $5.93 \pm 1.18$ | $100 \pm 0.00$ |
| DEFOG | $90.0 \pm 5.2$ | $0.6 \pm 2.3$ | $51.7 \pm 1.2$ | $55.6 \pm 73.9$ | $5.40 \pm 1.20$ | $90.0 \pm 5.1$ |
| DISCDIF | $0.00 \pm 0.00$ | $1.82 \pm 0.88$ | $86.38 \pm 6.37$ | $125.7 \pm 3.3$ | $11.28 \pm 1.06$ | $100 \pm 0.00$ |
| NM-DD | $60.50 \pm 4.30$ | $4.38 \pm 1.65$ | $50.92 \pm 0.89$ | $52.9 \pm 6.3$ | $5.70 \pm 0.30$ | $100 \pm 0.00$ |
| DIRIFM | $46.00 \pm 4.36$ | $4.26 \pm 0.54$ | $53.04 \pm 1.05$ | $53.0 \pm 8.9$ | $5.02 \pm 1.14$ | $100 \pm 0.00$ |
| UNSIDE | $78.50 \pm 4.64$ | $1.74 \pm 0.60$ | $49.94 \pm 1.07$ | $52.1 \pm 1.1$ | $5.90 \pm 1.05$ | $100 \pm 0.00$ |

Table 12: Molecule generation `Qm9H` on generation task.

|  | VALID (%) | FCD | NSPDK ($10^3$) | UNIQUE (%) |
|---|---|---|---|---|
| TRAIN. SET | $98.90 \pm 0.05$ | $0.062 \pm 0.002$ | $0.121 \pm 0.016$ | $99.81 \pm 0.04$ |
| DIGRESS | $95.4 \pm 1.1$ |  |  | $97.6 \pm 0.4$ |
| DISCDIF | $22.29 \pm 0.62$ | $4.246 \pm 0.454$ | $41.932 \pm 0.741$ | $73.72 \pm 0.56$ |
| NM-DD | $97.97 \pm 0.07$ | $0.366 \pm 0.021$ | $1.149 \pm 0.047$ | $95.23 \pm 0.17$ |
| DIRIFM | $92.20 \pm 0.19$ | $0.356 \pm 0.029$ | $0.495 \pm 0.010$ | $97.53 \pm 0.13$ |
| UNSIDE | $98.87 \pm 0.07$ | $0.152 \pm 0.011$ | $0.487 \pm 0.068$ | $96.33 \pm 0.14$ |

Table 13: Molecular generation on `Zinc250K` results.

|  | VALID (%) | FCD | NSPDK ($10^3$) | UNIQUE (%) | NOVEL (%) |
|---|---|---|---|---|---|
| TRAIN. SET | $100.00 \pm 0.00$ | $1.128 \pm 0.009$ | $0.10 \pm 0.00$ | $99.97 \pm 0.02$ | - |
| DIGRESS | $94.99$ | $3.482$ | $2.1$ |  |  |
| GRUM | $98.65$ | $2.257$ | $1.5$ |  |  |
| SID | $99.50 \pm 0.06$ | $2.06 \pm 0.05$ | $2.01 \pm 0.01$ | $99.84 \pm 0.02$ | $99.97 \pm 0.00$ |
| DISCDIF | $74.17 \pm 0.65$ | $4.78 \pm 0.14$ | $4.08 \pm 0.13$ | $100.00 \pm 0.00$ | $100.00 \pm 0.00$ |
| NM-DD | $99.92 \pm 0.01$ | $2.65 \pm 0.06$ | $3.48 \pm 0.06$ | $99.88 \pm 0.04$ | $99.95 \pm 0.03$ |
| DIRIFM | $97.32 \pm 0.12$ | $2.79 \pm 0.01$ | $1.92 \pm 0.07$ | $99.99 \pm 0.01$ | $99.99 \pm 0.01$ |
| OURS | $99.98 \pm 0.01$ | $1.79 \pm 0.03$ | $1.00 \pm 0.06$ | $100.00 \pm 0.00$ | $99.97 \pm 0.01$ |

## F  VISUALIZATIONS

We present here some visualizations of the generated and reference graphs.

Figure 6: `Qm9H`

Figure 7: `ZINC250K`

Figure 8: `Planar`

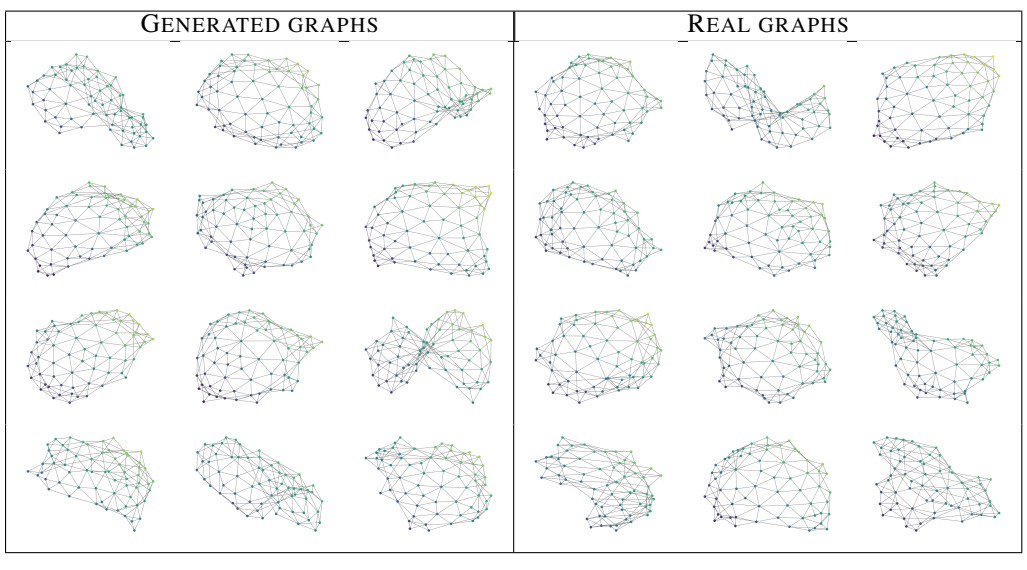

Figure 9: `Stochstic Block Model`

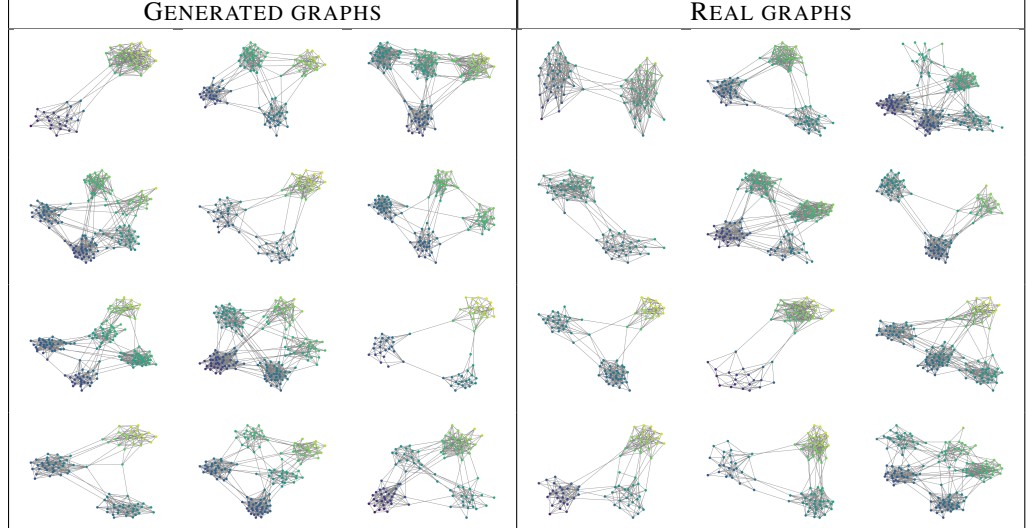

