# OpenReview forum: "Unrestrained Simplex Denoising for Discrete Data. A Non-Markovian Approach Applied to Graph Generation"
_ICLR.cc/2026/Conference — Submitted to ICLR 2026_

### Official Review · Reviewer_XHMC · 2025-10-27

**Soundness:** 2
**Presentation:** 3
**Contribution:** 2
**Rating:** 4
**Confidence:** 4

**Summary:**

The paper presents a diffusion model for graph generation based on a denoising process on the probability simplex. While previous simplex-based diffusion models for discrete data modeling are based on a Markovian process, the paper uses a non-Markovian noising that assumes independence across noisy states, which eliminates unnecessary constraints. Experimental results on graph generation tasks show improvements in small 2D molecule generation while being on par with SOTA baselines in unattributed graph generation.

**Strengths:**

- The writing is easy to follow and related works cover most of the relevant works.

- Application of n-dimensional simplex diffusion for graph generation seems to be a novel approach, to the best of my knowledge.

- The proposed method shows improvement in 2D molecule generation tasks.

**Weaknesses:**

- Why is the dependency across noisy states in Markovian diffusion considered unnecessary? This dependency is simply a byproduct of the Markovian design, which is not inherently necessary or unnecessary. Designing a non-Markovian process could potentially lead to better performance, though the reason for this improvement is not clearly explained. However, this alone does not imply that dependency across noisy states is unnecessary.

- Why is the proposed simplex-based diffusion model better than discrete diffusion models? The motivation in Figure 1 seems to apply to flow matching on simplex, and unclear how this connects with discrete diffusion models. Also, Dirichlet diffusion may show different results in the experiments in Figure 1.

- The experiments are limited to small-scale graph benchmarks: 2D molecules have at most 38 nodes, and unattributed graphs have at most 187 nodes (SBM). Is the method scalable to graphs with a larger number of nodes? This may be validated on the synthetic grid graphs, where controlling the number of nodes is possible and has been used in the baseline like Grum.

- The main claim of the paper is not validated sufficiently. The paper claims non-Markovian noising removes unnecessary constraints and improves performance, but results on unattributed synthetic graphs do not show this. Is the main claim wrong?

**Questions:**

Please address the questions in the weakness section.

---

> ### Author Response · Authors · 2025-11-17
>
> We thank the reviewer for his time and valuable feedback. We sincerely appreciate the comments, suggestions, and insights provided. We have substantially revised and improved the manuscript, and we are confident that the revised version is significantly clearer and stronger thanks to your contributions.
>
> ## 1. Advantage of our method
>
> At the beginning of Section 2.3, our intention is to clarify that, in statistics and physics, the strict definition of a diffusion process entails the Markov property (We clarified the formulation). We then immediately acknowledge that, in deep generative modeling, the term “diffusion’’ is used in a broader and more informal sense to describe a wide range of forward noising mechanisms. In this regard, we fully agree with the reviewer: the Markov property is a modeling \emph{choice}, not an inherent requirement, in generative modeling.
>
> We have added a full subsection 4.4 on the advantage of our method over standard diffusion and flow matching.  we specifically added a full paragraph on the reason why we think the non-Markovian apporach works better. We reproduce it here:
>
> Intuitively, diffusion and flow matching update the sample by interpolating between the current noisy state and the denoiser’s prediction, whereas our method directly resamples a less noisy state conditioned on the denoiser’s prediction.
> This observation provides intuition for the sampling benefits of our approach. In diffusion and flow matching, if $x_t$ lies in a region where the learned denoiser $P^\theta_{1\mid t}$ poorly approximates the true posterior $P_{1\mid t}$, or in a region highly sensitive to small perturbations (e.g., near equilibrium points), then the next iterate $x_{t+dt}$ remains close to $x_t$, causing approximation errors to compound over time. In contrast, our method resamples from $P^\theta_{1 \mid t}$ at each step, so $x_{t+dt}$ need not stay in the same local region. This reduces error compounding by allowing the trajectory to escape regions that are poorly approximated and/or sensitive to small perturbations.
>
> ##  2. Rationale for Continuous Denoising on the Simplex
>
> 2a. We thank the reviewer for this question. To clarify the distinction, we added a dedicated section: Rationale for Denoising on the Simplex (2.5). Below we reproduce the paragraph that specifically contrasts discrete diffusion with continuous diffusion on the simplex. (Please, see the full section for the difference between method operating on the simplex and methods operating on $R^K$.)
>
> Discrete and continuous noising differ fundamentally in the structure of the corrupted data. In discrete schemes, noisy samples remain sparse, whereas continuous perturbations produce fully connected weighted graphs. Continuous noising avoids discontinuities, most notably graph disconnection, and yields edge weights that naturally encode proximity, providing each node with information about its relative position to all others. More generally, continuous vectors are richer than one-hot representations of the same dimension and supply unique identifiers for nodes and edges, thereby alleviating the expressivity limitations of GNNs (Loukas, 2025). This prevent the need for costly, hand-crafted structural features typically required in discrete diffusion and flow matching methods (see Appendix C.2). Importantly, the sparsity of discrete noising does not translate into sparse GNN computation. Edge representations and probabilities must still be evaluated for all node pairs. Thus, the dense complexity remains.
>
>
> 2b. Regarding Figure 1, we agree that it illustrates only the difference between a linear–interpolant noising scheme and an explicit probability-path construction. It is not intended to convey a general principle beyond this point. Its unique purpose is to visualize one specific component of our method.
>
> ## 3. Scalability
>
> We would first like to emphasize that, aside from methods explicitly designed to address scalability, relatively few works report experiments on graphs larger than those in the SBM benchmark (although Grum is an exception). That said, we fully acknowledge that computational efficiency is an important concern. Addressing this point requires additional experiments, and given our current compute and time constraints, we are evaluating the best strategy to proceed. We will follow up once we have made progress on this.

---

> > ### Author Response · Authors · 2025-11-17
> >
> > ## 4. Empirical Validation of our Claim
> >
> > On this last point, we respectfully disagree with the reviewer.
> >
> > First, when using the *same denoiser* but *different sampling strategies* (our unrestrained method versus the constrained flow-matching variant reported as DiriFM) our approach consistently yields clear improvements, often by a substantial margin, across all metrics and all datasets except one (Spectral MMD on SBM). This ablation is the most rigorous way to isolate the effect of the sampling strategy, as it controls for all other factors. We believe that this result alone validates our claim.
> >
> > Second, to facilitate interpretation, we additionally report metrics computed between test-set graphs and graphs sampled from the training set. Overall, our method outperforms all baselines on 3 of the 4 datasets. On unattributed graphs, our model places 7 out of 10 metrics within the training-set error margin, more than any other baseline. In particular, on the Planar dataset (unattributed graphs), our method achieves the strongest performance.
> >
> > We acknowledge that on the SBM dataset our results are not clearly better compared to some baseline. However, unlike the controlled comparison with DiriFM (where the denoiser is identical), comparisons with those baselines may be influenced by numerous factors including architecture choices, training hyperparameters or the noise schedule, and therefore do not invalidate our claim.
> >
> > Finally, irrespective of the specific behavior on SBM, we emphasize that our method constitutes a significant conceptual and practical simplification over standard denoising approaches, while delivering strong empirical performance on most benchmarks. In particular, on molecular datasets we nearly resolve the long-standing validity issue. For these reasons, we are confident that our results are of broad relevance to the graph and molecular generative modeling communities.
> >
> >
> > We sincerely hope that we have addressed all of your concerns. Please let us know if you have any remaining questions, suggestions, or additional feedback—we would be very grateful to hear your perspective.
> > Thank you again for your time, thoughtful comments, and interest in our work.

---

> > > ### Author Response · Authors · 2025-11-23
> > >
> > > ## Scalability
> > >
> > > As suggested by the reviewer, we evaluate the computational efficiency of our model. Rather than testing on the _Grid_ dataset—which is saturated and has graphs with only two degrees of freedom—we report the more informative metrics of sampling time and parameter count. Our results show that our model is both faster and uses fewer parameters than competing approaches.
> > >
> > > ## Empirical Validation
> > >
> > > We have also rerun the experiments on the unattributed datasets for DiGress and Grum. Our motivation for doing so is as follows:
> > >
> > >
> > > On unattributed graph datasets such as _Planar_ and _SBM_, the test sets are relatively small, and sampling-based metrics exhibit high variance across runs. Moreover, evaluating many models, checkpoints, or seeds increases the risk of overfitting to the test set. Multiple sampling runs are therefore strongly recommended, and the corresponding variability should be assessed via standard deviations.
> > > Several baselines report results from a _single_ sampling run, including Grum, DiGress, and GraphBFN. These reported results even outperform samples drawn directly from the training set, indicating clear overfitting. For this reason, we reran the experiments for Grum and DiGress. Unfortunately, the official GraphBFN repository is currently empty, preventing us from reproducing that baseline.
> > > We describe in detail the procedure in Appendix D.6.
> > >
> > > With these updated results, our method is clearly superior across all datasets.
> > >
> > > We thank the reviewer for this suggestion, which highlights additional strengths of our method.
> > >
> > > In response to the other reviewers, we have updated the manuscript with additional experiments. We hope that these revisions address all of your concerns, and we look forward to any further questions or comments you may have.

---

> > > > ### Author Response · Authors · 2025-11-28
> > > >
> > > > Dear Reviewer XHMC,
> > > >
> > > > We have carefully addressed your remarks and questions, and we have substantially improved the manuscript accordingly. We are convinced that the revised version of our paper is significantly stronger thanks to your valuable feedback.
> > > >
> > > > We would be very happy to hear any additional comments you may have or to clarify any remaining questions.
> > > >
> > > > Thank you again for your time and constructive input.
> > > >
> > > > Sincerely,
> > > > The Authors

---

### Official Review · Reviewer_jzMT · 2025-10-30

**Soundness:** 3
**Presentation:** 3
**Contribution:** 3
**Rating:** 4
**Confidence:** 4

**Summary:**

The paper is clearly written. The formulations for both the reverse and forward processes are very clean. Figure 1 clearly illustrates the comparison with linear interpolation. The experimental results are complete. Overall, I find the paper sound and complete.

**Strengths:**

The paper is clearly written. The formulations for the reverse process and the forward process are very clean. The formulas are easy to follow and convey the authors’ ideas. Figure 1 clearly illustrates the comparison with linear interpolation, as commonly done in discrete flow matching. I find the paper sound and complete.

**Weaknesses:**

1. Figure 1 clearly illustrates the disadvantages of previous discrete flow matching formulations. However, even with the thorough discussion of related work in Section 2.2, it is still difficult for me to clearly understand the differences or relationships between the proposed method and (1) Dirichlet diffusion, and (2) Dirichlet flow matching in spirit. Could the authors explain explicitly on this point? This will make the contribution of the paper more clear. At the same time, compared to the other group of methods that maps the prediction back to the simplex, what is the intuition that Dirichlet diffusion may function better?
2. There is related work that could be included in the comparison or discussed about, such as Graph BFN (Smooth Interpolation for Improved Discrete Graph Generative Models), which also performs denoising within the simplex space.
3. Additional ablations could further support the method: for example, does the noise schedule matter, or do any hyperparameters require tuning for different datasets?

**Questions:**

1. In line 346, the authors mention: “but we can also approximate this prior using a model trained with the probability path…”
Could the authors explicitly explain how this approximation is achieved without specifying the prior directly?
2. Another point of interest in the current research direction is generating graphs with fewer sampling steps. I wonder whether the authors have explored reducing the number of steps.

---

> ### Author Response · Authors · 2025-11-17
>
> We thank the reviewer for his time and valuable feedback. We sincerely appreciate the comments, suggestions, and insights provided. We have substantially revised and improved the manuscript, and we are confident that the revised version is significantly clearer and stronger thanks to your contributions.
>
> ## 1a:  Differences with diffusion and flow matching in spirit.
>
> Intuitively, diffusion and flow matching update the sample by interpolating between the current noisy state and the denoiser’s prediction (optionallly followed by an additional noise injection). As a result, $x_{t+dt}$ remains in the immediate neighborhood of $x_t$. In contrast, our method directly resamples a less noisy state conditioned on the denoiser’s prediction, so $x_{t+dt}) is generally *not* constrained to lie near $x_t$.
>
> We added this explaination in Section 4.4. We also provide an intuition why this sampling method is beneficial, and why this represents an important simplification over both methods.
>
> ## 1b: Advantage of modeling on the simplex
>
> Thank you for question, we added a full paragraph specifically on this point on the Background section (Section 2). Here is our explaination:
>
> Modeling categorical distributions in $\mathbb{R}^k$ introduces limitations that are avoided by operating directly on the probability simplex. First, points on the simplex have only k-1 degrees of freedom; embedding them in $\\mathbb{R}^k$ adds a redundant dimension that carries no information and unnecessarily complicates the dynamics. Second, simplex-based noising naturally defines explicit and interpretable probability paths over categorical distributions (Section 3), whereas Euclidean noising requires nonlinear projection or renormalization steps whose behavior is harder to characterize. Finally, points on the simplex admit a direct probabilistic interpretation, which is both conceptually elegant and facilitates the design of the denoising process (Section 4.2).
>
> ## 2. Additional Reference
>
> We gratefully thanks the reviwer for this important reference that we missed. We modify the related work section and specifically the paragraph about graph generative models operating on the simplex including now a discussion about it. We also added the reference to the baselines. This does not change our conclusion.
>
> ## 3 Ablation study
> We emphasize that the results section already includes a dedicated ablation study examining the sampling method, the choice of prior, and the number of sampling steps (see also our response to Question 2). Unfortunately, many of these results had to be moved to the appendix due to page limitations.
>
> We appreciate the reviewer suggestion and we will add a specific ablation about the noise schedule. As it requires to retrain models, this requires more time and will be added later. We will come back to you, when it is done.
>
> ## Questions:
>
> Question 1:
> We explain this directly on the following of the same paragraph.
> We leverage the fact that we have: dimension-wise:
> $E_{x_1} [ q_t^\star(x_t \mid x_1)] = q^{marg}_0(x_0)
> \iff \alpha_t = \kappa$, and the approximation in Equation 8.
> We underscore that, at inference, the prior is explicitly defined as $q_0^{marg}$.
> Please, let us know if you need more explanation on this point.
>
> Question 2:
> We did ablate the effect of the number of function evaluation (NFE), the results are provided in Figure 3. For completeness, we have added the corresponding table in appendix (Table 12). For example, we show that we reach state-of-the-art validity on Zinc250k in less than 10% of the steps.
>
> We sincerely hope that we have addressed all of your concerns. Please let us know if you have any remaining questions, suggestions, or additional feedback. We would be very grateful to hear your perspective.
> Thank you again for your time, thoughtful comments, and interest in our work.

---

> > ### Author Response · Authors · 2025-11-23
> >
> > ## 3.Ablation Study
> >
> > As suggested, we have added an ablation study on the noise scheduler. The full description and complete results are provided in Section E.2.2.
> > We summarize the additional experiment as follow:
> >
> > We assess the effect of the noise scheduler $\alpha_t$, which parametrizes our probability paths
> > $ \text{Dir}\bigl(x_t; 1 + \alpha_t x_1 \bigr) $, with $\alpha_t = -a\log(1-t)$
> > Concretely, we evaluate our model across a range of values for the hyperparameter $a$.
> > We draw two main conclusions from this experiment:
> > (1) the Voronoi-probability analysis provides a reliable tool for calibrating the noise scheduler, and
> > (2) our model is robust to a wide range of scheduler choices.
> >
> > We thank the reviewer for this suggestion, which highlights an additional strength of our method.
> >
> > In response to the other reviewers, we have updated the manuscript with additional experiments. We hope that these revisions address all of your concerns, and we look forward to any further questions or comments you may have.

---

> > > ### Comment · Reviewer_jzMT · 2025-11-27
> > >
> > > Thank the reply from the authors.
> > > I will raise the rating to 6.

---

### Official Review · Reviewer_nfL1 · 2025-10-31

**Soundness:** 3
**Presentation:** 3
**Contribution:** 2
**Rating:** 6
**Confidence:** 3

**Summary:**

The paper proposes a diffusion-based method for generating graphs with categorical attributes. Noised versions of categorical variables (node and edge attributes) are represented as points on a simplex, and the denoising process operates in this space. The motivation is to design a diffusion process that avoids the discontinuities of discrete noising. The forward noising is non-Markovian, i.e., noised samples are independent and depend only on the clean data. The method is evaluated on molecular graphs and synthetic (SBM, planar graph) datasets.

**Strengths:**

- Generative modeling for discrete data is a relevant topic. The idea of applying non-Markovian diffusion to graph generation is conceptually interesting.
- The authors provide a clear theoretical exposition, especially related to noising on the simplex.
- In general, the paper is clearly written and easy to follow. The related work is presented well.
- The experimental results are solid and show that relaxing the Markov assumption doesn't seem to harm empirical performance.

**Weaknesses:**

The downside of denoising in the simplex space is that graph sparsity is lost: the communication graph inevitably becomes fully connected. This is not explicitly discussed in the paper, though there is a related work section on scalable methods.

The authors note that simplex-based diffusion for graph generation has already been done in (Liu et al. 2025). The use of Dirichlet distributions for categorical variables appears to be a relatively straightforward generalization of the earlier Beta-distribution based approach in Liu et al.

Likewise, the challenges related to noising on the simplex are already studied in Stark et al. (2024). The proposed Voronoi-based probability construction is theoretically sound, but it does not by itself represent a major conceptual advance.

**Questions:**

The use of non-Markovian noise is interesting, though the theoretical analysis in Section 4.2 would benefit from clearer references to related work on this topic to highlight which aspects are novel.

**Typos and grammar:**
- l.16: "yet most approaches either operate
directly in the discrete state space, causing abrupt state changes and discontinuities." --> remove either
- l.234: the definition of L would be useful here
- l.273 $Cat(\pi)$ undefined
- l.303: duplicate "univariate case"
- l.306: "Let assume" --> Assume
- l.365: "... framework supports both. Because ..." --> replace period with comma
- l.957: The denoisers are Graph Neural Network,  --> Networks
- l.981: $W^lsrc, W^ltrg, and W^ledge$   --> $W^l_{src}$; other similar errors occur in this section

---

> ### Author Response · Authors · 2025-11-17
>
> We thank the reviewer for his time and valuable feedback. We sincerely appreciate the comments, suggestions, and insights provided. We have substantially revised and improved the manuscript, and we are confident that the revised version is significantly clearer and stronger thanks to your contributions.
>
> ## 1. Downside of denoising on the simplex
>
> In the revised manuscript, we added an extended paragraph on this point as well as a full complementary section in the appendix . We respectfully disagree with the concern that the fully connected graphs produced by simplex noising represent a drawback. As explained in the dedicated paragraph of the revision, which we reproduce here, the dense structure is in fact beneficial for several reasons:
>
> Discrete and continuous noising differ fundamentally in the structure of the corrupted data. In discrete schemes, noisy samples remain sparse, whereas continuous perturbations produce fully connected weighted graphs. Continuous noising avoids discontinuities, most notably graph disconnection, and yields edge weights that naturally encode proximity, providing each node with information about its relative position to all others. More generally, continuous vectors are richer than one-hot representations of the same dimension and supply unique identifiers for nodes and edges, thereby alleviating the expressivity limitations of GNNs (Loukas, 2019). This prevent the need for costly, hand-crafted structural features typically required in discrete diffusion and flow matching methods (see Appendix C.2). Importantly, the sparsity of discrete noising does not translate into sparse GNN computation. Edge representations and probabilities must still be evaluated for all node pairs. Thus, the dense complexity remains.
>
> ## 2. Generalization from the Beta distribution:
>
> We have updated the related-work section to clarify this point, particularly in our discussion of Beta Diffusion. In brief, a direct extension from the Beta distribution on the
> 1-simplex to a Dirichlet-based diffusion or flow-matching process on the
> K-simplex has proven difficult in practice. Several works, including our own implementation of the Dirichlet Flow Matching baseline, attempted such generalizations but obtained unsatisfactory results. Where these approaches struggle, our denoising formulation provides a simple yet effective alternative.
>
> For completeness, the revised paragraph on Beta Diffusion now reads as follows:
>
> Beta diffusion (BetaGraph) applies continuous noise on the $1$-simplex via the Beta distribution. An extension to the $k$-dimensional probability simplex might appear straightforward. However, no model achieves this extension directly, indicating that this is more challenging than it appears. (BetaGraph} generalize their approach by encoding $k$-categorical variables as $k$ binary variables. (CatFlow) attempt to adopt the Dirichlet flow matching formulation but report poor performance, leading them to relax the noising process to $\mathbb{R}^k$. Recently, Graph Bayesian Flow Network (graphBFN) proposes a diffusion mechanism that adds Gaussian noise in $\mathbb{R}^k$
> followed by a projection into the simplex. Sampling proceeds via a dual update scheme in which the update type is selected based on the Euclidean progress achieved by the previous step.
> Taken together, these developments suggest that simplex-based denoising is more difficult than one might expect. We address this challenge by relaxing the standard Markovian assumptions used in diffusion and flow matching, yielding a simple yet effective model that operates _directly_ on the probability simplex.

---

> ### Author Response · Authors · 2025-11-17
>
> ## 3. Contribution
>
> We fully agree that the Voronoi-based probability construction is not, on its own, a major conceptual advance. Our primary contributions are those presented in Section 4, and we elaborate on them below in response to your question.
>
>
> ## Response to the questions
>
> We made explicit in the revised manuscript that SID (Boget, 2025) can be viewed as the discrete analogue of our method. To clarify the relationship, we added a detailed, side-by-side comparison in the appendix (C.3), identifying both the shared components and the elements that represent strict contributions of our work. For completeness, we reproduce this comparison here:
>
> Our Unside framework can be viewed as a continuous analogue of \emph{Simple Iterative Denoising} (SID). Below we outline the key similarities and differences.
>
> _Noising_:
> In SID, the corruption process follows discrete diffusion for categorical data (Austin), where intermediate noisy distributions linearly interpolate between the data distribution and a stationary prior.
> As discussed in Section 3, this interpolation is not suitable in the continuous setting on the simplex. Accordingly, Unside adopts an explicit probability path within the simplex.
>
> _Independence Assumption_:
> Both SID and Unside rely on the same conditional-independence assumption (Assumption 4.1), which is central to the formulation.
>
> _Denoising_:
> The denoising rules in Unside follow directly from the conditional-independence assumption and are closely related in spirit to SID, even if Equation 5 does not appear in SID explicitly. A notable distinction is that our method couples a _continuous_ kernel $q_t$ on the simplex with a _discrete_ posterior $P_{1\mid t}$, an original hybrid construction that is part of our contribution.
>
> _Convergence_:
> Proposition 4.3 and its proof are original to this work. SID provides no analogous convergence result. We regard this result as a core theoretical contribution.
>
> _Advantage over Diffusion and Flow Matching_:
> Although there are conceptual similarities at a high level—including the connection to compounding denoising errors, the underlying intuition differs substantially between the continuous and discrete settings. In particular, the observation that $x_{t+dt}$ remains close to
> $x_t$, and the reasoning that follows from this, holds only in the continuous case and does not translate directly to discrete models.
>
> _Parametrization and learning_:
> Both approaches employ standard parameterizations and training objectives common to discrete diffusion and flow-matching methods.
>
> _Probability paths and priors_:
> Our choices and analysis of probability paths and priors are specific to continuous simplex noise and have no direct counterpart in SID.
>
> _Guidance_:
> As explained in the main text, guidance in Unside is obtained via direct adaptations of standard diffusion techniques (classifier and classifier-free guidance). SID does not include a guidance mechanism.
>
> Beyond this element-wise comparison, our primary contribution is to integrate these components into a simple and efficient framework for graph generation that achieves state-of-the-art performance across multiple datasets.
>
> ## 4. Typos and Grammar
>
> Thank you for your careful reading. It is very much appreciated.
>
> We sincerely hope that we have addressed all of your concerns. Please let us know if you have any remaining questions, suggestions, or additional feedback. We would be very grateful to hear your perspective.
> Thank you again for your time, thoughtful comments, and interest in our work.

---

> > ### Author Response · Authors · 2025-11-23
> >
> > In response to the other reviewers, we have updated the manuscript with additional experiments. We hope that these revisions address all of your concerns, and we look forward to any further questions or comments you may have.

---

> > > ### Author Response · Authors · 2025-11-28
> > >
> > > Dear Reviewer nfL1,
> > >
> > > We have carefully addressed your remarks and questions, and we have substantially improved the manuscript accordingly. We are convinced that the revised version of our paper is significantly stronger thanks to your valuable feedback.
> > >
> > > We would be very happy to hear any additional comments you may have or to clarify any remaining questions.
> > >
> > > Thank you again for your time and constructive input.
> > >
> > > Sincerely,
> > > The Authors

---

### Author Response · Authors · 2025-11-17
**List of changes**

First, we thank all reviewers for their time and valuable feedback.

We have substantially revised the manuscript with two goals in mind: (i) to clarify the contributions and novelty of our work, and (ii) to better motivate our methodological choices. Importantly, we emphasize that the theoretical simplicity of our approach should not be mistaken for a lack of contribution. On the contrary, we view this simplicity as a core strength and a central contribution of the paper.

Below, we summarize the main changes made in the revised manuscript:

- **Clarified contributions.** We reformulated and reordered the list of contributions to highlight the simplification our method provides relative to diffusion and flow matching, as well as the theoretical guarantees we establish.

- **Expanded background and motivation.** We added a new subsection in the *Background* section to motivate our approach in comparison to discrete denoising methods and continuous methods that do not operate on the simplex. We also added a complementary appendix section analyzing denoiser expressivity in discrete vs. continuous settings.

- **Clearer connection to SID.** We revised the beginning of the *Method* section to clarify the relationship between our approach and SID, and we added a detailed comparison in the appendix to delineate shared elements and our own contributions.

- **New subsection on differences from diffusion/flow matching.** We added a subsection explaining how our method differs from diffusion and flow matching and why these differences are beneficial.

- **Updated related work and baselines.** We incorporated the references suggested by the reviewer and added the corresponding baseline.

- **Improved baseline interpretation.** We now report metrics comparing test graphs to samples from the training set, and we highlight all results within the corresponding error margin, as we consider them equally strong.

- **Improved experimental results.** We reran several experiments using larger networks (matching the architectures used by other baselines), leading to improved metrics. We updated the technical details accordingly.

- **Qm9H dataset clarification.** We removed DeFog from the Qm9H baselines after realizing it uses a dataset variant with aromatic bonds, whereas we use the kekulized version. Since results are not directly comparable, we added a dedicated appendix section specifying the version and evaluation protocol we use.

- **Minor corrections.** We fixed typos and improved other stylistic and formatting details.

We sincerely hope that we have addressed all of your concerns.
Thank you again for your time, thoughtful comments, and interest in our work.

---

> ### Author Response · Authors · 2025-11-23
> **List of changes 2**
>
> We have now conducted additional experiments. We summarize the additional changes made in the revised manuscript as follow:
>
> - **Effectiveness of our method**: We include an additional experiment to demonstrate the effectiveness of our method (Section **E.1**).
>   In particular, we show that our sampling procedure is **faster** than both **Grum** and **DiGress**, while achieving **higher performance** with **fewer parameters** (see Section **E.1**).
>
> - **Robustness to scheduler variations**: We perform an ablation study on the scheduler parameter $a$, which governs the Voronoi probabilities as described in Section **3.2** (Section **E.2.2**).
>   The experiment supports two main conclusions:
>
>   1. The Voronoi-probability analysis provides a reliable mechanism for calibrating the noise scheduler.
>   2. Our model remains robust across a wide range of scheduler configurations.
>      Full experimental results and discussion can be found in Section **E.2.2**.
>
> - **Fair Comparison**:
> To ensure a fair comparison, we rerun the unattributed-graph experiments for **DiGress** and **Grum**, reporting averages over five independent runs along with the corresponding standard deviations. The motivation for this procedure is provided in Appendix **D.6**. With these updated results, our method is clearly superior across all datasets.
>
> - **Minor adjustments**: were made in the main text to reference this additional material.
>
> Thank you again for your time, thoughtful comments, and interest in our work.

---

### Comment · Area_Chair_9Woj · 2025-11-23
**Author Rebuttal Available - Please Review & Discuss**

Dear Reviewers,

The authors have submitted their rebuttal addressing your reviews. Please take the time to:

1. Read the rebuttal carefully
2. Ask clarifying questions if anything remains unclear
3. Update your scores and reviews based on the authors' responses

Please be mindful of timing: If you have follow-up questions for the authors, **post them early enough to give them adequate time to respond** before the discussion period closes on December 3rd.

Your timely engagement is crucial for a fair and thorough review process.

Thank you for your continued effort on this paper.

Best regards,
Area Chair

---

### Author Response · Authors · 2025-12-01
**Summary**

**Dear Area Chair,**

To facilitate your evaluation, we provide a summary of the main contributions of our paper and the improvements made during the rebuttal phase.

## **Main Contributions**

* **A new denoising paradigm for graph generation.**
  Our method introduces a *new sampling mechanism* specifically designed for categorical data and graphs. Unlike most generative model for graphs, it is not an adaptation of an existing framework.

* **Substantial simplification over diffusion and flow matching.**
We emphasize that the simplicity of our approach represents *a major contribution* rather than a weakness, as it leads to improved empirical performance. While theoretically simple, our approach removes key constraints inherent to diffusion and flow-matching models.

* **Resolution of key limitations of prior methods.**
  Our approach mitigates *compounding denoising errors* (Section 4.4) and, by operating directly on the $K-1$-simplex, offers advantages over models in $\mathbb{R}^K$ and discrete space (Section 2.5).

* **Theoretical guarantees.**
  We prove convergence of our denoising process to the data distribution (Section 4.3) under the true denoising distribution $p_{1|t}(x \mid x_t)$.

* **Strong empirical performance.**
  Across synthetic and molecular graph benchmarks:

  * Our method surpasses strong discrete diffusion and flow-matching baselines, achieving new SOTA results on multiple datasets.
  * Sampling is faster and requires fewer parameters than competing models.
  * Competitive results are achieved with tens of NFEs, while baselines require hundreds.
  * Performance is robust across a wide range of noise schedules.

## **Improvements Made After Rebuttal**

We acknowledge that the initial manuscript did not sufficiently highlight the motivation and contributions. We have substantially revised the paper to make these aspects clearer:

* Reworked the contribution summary in the Introduction.
* Added **Section 2.5** (Rationale for denoising on the simplex), complemented by **Appendix C.2**.
* Expanded the introduction of **Section 4** to clarify the relationship with SID (Boget, 2025), with a full comparison in **Appendix C.3**.
* Added **Section 4.4** (Advantages over diffusion and flow matching).

We also added new experiments, including:

* **Sampling time and parameter count comparison** (Appendix E.1).
* **Scheduler ablation study** (Appendix E.2.2).

## **Final Remarks**

Some reviewers provided limited interaction and sometimes overlooked parts of the manuscript (e.g., suggesting NFE reductions analysis despite dedicated figure already included). We believe our paper represents an important contribution for both **its conceptual simplicity** and **its strong empirical improvements across multiple benchmarks**. We are convinced it constitutes a substantial contribution for the scientific communities working on graph and generative modeling.

We sincerely thank you for your time and consideration.

---

### Meta-Review · Area_Chair_wmec · 2026-01-02

**Summary:**

## Summary of the paper

This paper proposes Unrestrained Simplex Denoising (UNSIDE), a non-Markovian denoising framework for discrete data that operates directly on the probability simplex. By assuming conditional independence across noisy states (given the clean data), the method removes the continuity constraints imposed by diffusion and flow-matching models. The authors provide a clean theoretical formulation, including convergence guarantees under the true denoising posterior, and introduce Voronoi-probability analysis to design and calibrate probability paths on the simplex. Empirically, the approach achieves strong or state-of-the-art results on molecular and synthetic graph generation benchmarks, with improved sampling efficiency and robustness.

## Main reviewer concerns before rebuttal

Across reviewers, the main concerns were:

- Novelty and positioning: whether the method is a substantial advance beyond prior simplex-based diffusion/flow-matching (e.g., Dirichlet diffusion, SID), or mainly a simplification/reformulation.

- Justification of non-Markovian design: why removing Markovian dependence is principled rather than a modeling choice, and whether it truly addresses compounding errors.

- Clarity of comparison to related work: especially Dirichlet diffusion, Dirichlet flow matching, SID, and GraphBFN.

- Empirical scope and validation: limited scale of graph benchmarks, mixed gains on SBM, and lack of some ablations (noise schedule, sampling efficiency).

- Scalability and sparsity concerns: dense simplex representations vs. sparse graphs.

## How the rebuttal addressed these concerns

The rebuttal substantially strengthened the submission by:

- Clarifying novelty and relationships: adding explicit comparisons to SID and other simplex-based methods, and clearly framing UN­SIDE as a continuous analogue with original convergence results.

- Expanding empirical evidence: adding ablations on noise scheduling, sampling steps (NFE), parameter count, and wall-clock efficiency, as well as fair multi-run evaluations of baselines.

- Addressing sparsity/scalability concerns: clarifying that dense simplex representations do not increase asymptotic GNN cost and demonstrating faster sampling with fewer parameters.

Overall, most factual and empirical concerns were directly addressed. The remaining reservations are twofold: (1) they primarily concern perceived conceptual novelty and large-scale scalability, rather than technical correctness; and (2) the paper still does not clearly justify the core motivation for the key technical design choice—namely, adopting a non-Markovian noising/denoising formulation.

In particular, Reviewer XHMC’s question about why the Markovian dependency is “unnecessary,” and what direct advantage the non-Markovian assumption provides, goes to the heart of the paper’s contribution. While the rebuttal clarifies how the proposed procedure differs from diffusion/flow matching (e.g., resampling versus local interpolation) and argues that this may reduce compounding errors, it stops short of providing a concrete, direct justification that convincingly explains when and why the non-Markovian scheme is preferable beyond being an alternative modeling choice.

As a result, although I believe the approach is promising and the overall idea could develop into an interesting contribution, the incomplete motivation/justification of the central design decision leads me to lean toward rejection at this stage.

**Reviewer Concerns:**

See summary above.

**Reviewer Scores:**

Reviewer jzMT: explicitly raised the rating to 6 after the rebuttal (confirmed).

Reviewer nfL1: originally marginally positive at 6; would likely remain at 6.

Reviewer XHMC: remained skeptical (4) about the necessity of the non-Markovian design and the strength of empirical evidence; would likely remain at 4.

---

### Decision · Program_Chairs · 2026-01-26

Reject